# PRISM: Enhancing PRotein Inverse Folding through Fine-Grained Retrieval on Structure-Sequence Multimodal Representations

**Sazan Mahbub**$^{c}$**, Souvik Kundu**$^{i}$**, Eric P. Xing**$^{c,m,g}$
$^{c}$Carnegie Mellon University, $^{m}$Mohamed bin Zayed Univeristy of AI, $^{g}$GenBio AI, $^{i}$Intel

## Abstract

Designing protein sequences that fold into a target 3-D structure, termed as the inverse folding problem, is central to protein engineering. However, it remains challenging due to the vast sequence space and the importance of local structural constraints. Existing deep learning approaches achieve strong recovery rates, however, lack explicit mechanisms to reuse fine-grained structure-sequence patterns conserved across natural proteins. To mitigate this, we present PRISM, a multimodal retrieval-augmented generation framework for inverse folding. PRISM retrieves fine-grained representations of *potential motifs* from known proteins and integrates them with a hybrid self-cross attention decoder. PRISM is formulated as a latent-variable probabilistic model and implemented with an efficient approximation, combining theoretical grounding with practical scalability. Experiments across multiple benchmarks, including CATH-4.2, TS50, TS500, CAMEO 2022, and the PDB date split, demonstrate the fine-grained multimodal retrieval efficacy of PRISM in yielding SoTA perplexity and amino acid recovery, while also improving the foldability metrics (RMSD, TM-score, pLDDT).

## 1 Introduction

Designing protein sequences that fold into a prescribed three-dimensional structure—the *inverse folding problem*—is a long-standing challenge in computational biology with far-reaching implications in biophysics, enzyme engineering, and drug discovery. Unlike structure prediction, where methods such as AlphaFold2 (John et al., 2021) have achieved transformative success, inverse folding must contend with a vast combinatorial search space: many distinct amino acid sequences can realize the same structural fold, and subtle local variations often determine stability and function. This underdetermined nature has made inverse folding scientifically important and challenging.

Recent deep learning approaches have made significant progress (Tian et al., 2026). Autoregressive sequence generators such as ProteinMPNN (Dauparas et al., 2022) demonstrated strong sequence recovery and practical utility across monomers, oligomers, and designed nanoparticles. PiFold (Gao et al., 2022) combined expressive encoders with efficient decoders, offering substantial speedups while maintaining competitive accuracy. More recent works have exploited pretrained protein language models. LM-Design, DPLM, and DPLM-2 (Zheng et al., 2023; Wang et al., 2024a;b) leverage large-scale sequence modeling and diffusion-based generation, while AIDO.Protein (Sun et al., 2024) scaled it to billions of parameters using mixture-of-experts training. Despite these advances, current architectures remain limited: they lack explicit mechanisms to reuse fine-grained structure–sequence patterns (e.g., recurring motifs) that are evolutionarily conserved and central to protein function.

**Our key insight** is that inverse folding can benefit from an explicit *retrieval mechanism* that grounds predictions in the rich diversity of known proteins at a fine-grained level. By treating local structure–sequence neighborhoods as reusable building blocks, one can supplement end-to-end generative modeling with *memory-based context*. This motivates **PRISM**, a multimodal retrieval-augmented generation (RAG) framework that reframes inverse folding through explicit representation, retrieval, and attribution. Instead of relying solely on a monolithic encoder, PRISM retrieves embeddings of

---

Contact: smahbub@cs.cmu.edu, souvikk.kundu@intel.com, epxing@cs.cmu.edu

*potential motifs* from a vector database of proteins, and aggregates them with a hybrid transformer decoder to refine sequence emission. This introduces an *explicit inductive bias*: each residue prediction is guided by retrieved local fragments, while the hybrid decoder integrates these fragment-level priors with global backbone context.

Our major contributions are:

- *A retrieval-augmented framework.* We propose PRISM, the first RAG framework for protein inverse folding that operates at residue-level granularity, retrieving fine-grained multimodal representations for potential motifs and reusing conserved local patterns during sequence design.
- *A theoretically grounded formulation.* We derive a latent-variable model that factorizes representation, retrieval, attribution, and emission, and provide an efficient approximation for implementation, ensuring both theoretical soundness and computational efficiency.
- *Extensive empirical validation.* Through comprehensive experiments across five benchmarks and multiple evaluation metrics, we establish new state of the art in both sequence recovery and structural fidelity, while incurring only negligible runtime overhead. Detailed ablations validate the role of each design choice in our framework.

## 2 PRELIMINARIES

The *protein inverse folding problem* aims to design an amino acid sequence that is compatible with a given three-dimensional protein backbone. Formally, let a backbone structure be specified by atomic coordinates $B = (p_1, \ldots, p_n)$, where each $p_i \in \mathbb{R}^3$ denotes the position of the $i$-th backbone atom. The goal is to predict a sequence $\mathbf{S} = (S_1, \ldots, S_L)$, where each residue $S_j$ is drawn from the standard amino acid vocabulary $\mathcal{V}$. A inverse folding model thus learns the following distribution

$$P(\mathbf{S} \mid B) = \prod_{j=1}^{L} P(S_j \mid B),$$

which assigns probabilities to candidate sequences consistent with the target backbone. To represent protein structures, modern approaches often construct a residue-level graph $G = (V, E)$, where nodes $v_i \in V$ correspond to residues and edges $e_{ij} \in E$ capture spatial or physicochemical interactions (Dauparas et al., 2022; Alam et al., 2024; Gao et al., 2022; Mahbub & Bayzid, 2022). A model then encodes $G$ and outputs a distribution over residues for each position, either autoregressively (predicting residues sequentially) or non-autoregressively (predicting all positions in parallel). The designed sequence is obtained by sampling or decoding from this distribution. Detailed discussion on related work has been provided in Appendix A.

## 3 PRISM: A MULTIMODAL RAG FRAMEWORK FOR INVERSE FOLDING

We introduce PRISM, a multimodal retrieval-augmented generation (RAG) framework for protein inverse folding that operates at residue-level granularity. We first formalize fine-grained structural–sequential regularities via *motifs* and *potential motifs*, then derive a latent-variable model that factors retrieval, attribution, and emission. We conclude with concrete instantiations of the representation, vector database, retrieval kernel, and the training objective.

### 3.1 MOTIFS AND POTENTIAL MOTIFS

> **Definition 3.1** (Protein Motif). *It is a recurring local structural–sequential pattern of residues that is evolutionarily conserved and often functionally significant. Formally, it can be described as a short stretch of amino acids together with its surrounding 3-D conformation, capturing local folding rules and biochemical properties independent of the global protein context.*

> **Definition 3.2** (Potential Motif). *We generalize motifs by treating each residue together with its local 3-D neighborhood as a* potential motif. *A potential motif may or may not align with a canonical structural motif, but serves as a fine-grained* motif-like unit *that encodes transferable structure–sequence information. These representations are the building blocks for retrieval and sequence emission in our RAG framework.*

### 3.2 LATENT-VARIABLE FORMULATION

**Modeling Objective.** Given a target backbone 3-D structure $B$ and a fixed residue-level vector database $D$ (whose entries represent potential motifs in local neighborhoods), our goal is to model

the conditional distribution over amino-acid sequences $p(\mathbf{S} \mid B, D)$. Directly parameterizing this conditional is challenging due to combinatorial sequence space and long-range dependencies. We therefore introduce latent variables that capture *retrieval* of locally similar neighbors and their *attribution* to each site before emitting the final sequence.

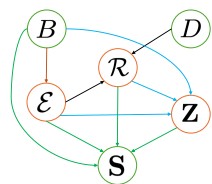

**Latents for representation, retrieval, and attribution.** Let $\mathcal{E} = \{\mathcal{E}_i\}_{i=1}^L$ denote latent variables for the *potential-motif representation*, and $\mathcal{R} = \{\mathcal{R}_i\}_{i=1}^L$ denote a latent retrieval hypothesis, where $\mathcal{R}_i$ are neighbors retrieved from $D$ for the (potential) motif in residue $i$'s locality (Fig. 2, Point ①; Sec. 3.4). We define the *retrieval kernel* as $p(\mathcal{R} \mid \mathcal{E}, B, D)$. Let $\mathbf{Z}$ denote *attribution* variables with conditional $p(\mathbf{Z} \mid \mathcal{R}, \mathcal{E}, B, D)$ that specifies how retrieved neighbors contribute to emissions $\mathbf{S} = \{S_i\}_{i=1}^L$, with $\mathbf{S} \sim p(\mathbf{S} \mid \mathbf{Z}, \mathcal{R}, \mathcal{E}, B, D)$.

Figure 1: Probabilistic graphical model of our proposed approach.

**Basic generative factorization.** The joint distribution factorizes as

$$p(\mathbf{S}, \mathcal{E}, \mathcal{R}, \mathbf{Z} \mid B, D) = \underbrace{p(\mathcal{E} \mid B, D)}_{\text{representation}} \underbrace{p(\mathcal{R} \mid \mathcal{E}, B, D)}_{\text{retrieval kernel}} \underbrace{p(\mathbf{Z} \mid \mathcal{R}, \mathcal{E}, B, D)}_{\text{attribution}} \underbrace{p(\mathbf{S} \mid \mathbf{Z}, \mathcal{R}, \mathcal{E}, B, D)}_{\text{sequence emission}}. \quad (1)$$

Using the conditional independences $\mathcal{E} \perp\!\!\!\perp D \mid B$, $\mathcal{R} \perp\!\!\!\perp B \mid \mathcal{E}$, and $\{\mathbf{Z}, \mathbf{S}\} \perp\!\!\!\perp D \mid \mathcal{R}$, we obtain

$$p(\mathbf{S}, \mathcal{E}, \mathcal{R}, \mathbf{Z} \mid B, D) = \underbrace{p(\mathcal{E} \mid B)}_{\text{representation}} \underbrace{p(\mathcal{R} \mid \mathcal{E}, D)}_{\text{retrieval kernel}} \underbrace{p(\mathbf{Z} \mid \mathcal{R}, \mathcal{E}, B)}_{\text{attribution}} \underbrace{p(\mathbf{S} \mid \mathbf{Z}, \mathcal{R}, \mathcal{E}, B)}_{\text{sequence emission}}. \quad (2)$$

Marginalizing over the latents yields

$$p(\mathbf{S} \mid B, D) = \mathbb{E}_{p(\mathcal{E} \mid B) \, p(\mathcal{R} \mid \mathcal{E}, D) \, p(\mathbf{Z} \mid \mathcal{R}, \mathcal{E}, B)} \big[ p(\mathbf{S} \mid \mathbf{Z}, \mathcal{R}, \mathcal{E}, B) \big]. \quad (3)$$

The corresponding probabilistic graphical model is shown in Fig. 1. This formulation induces a *family of valid objectives* arising under different approximations or parameterizations of the latents $\mathcal{E}$, $\mathcal{R}$, and $\mathbf{Z}$. We describe the model components and their efficient deterministic instantiations below. Fig. 2 depicts the overall pipeline of our proposed framework.

### 3.3 STRUCTURE–SEQUENCE MULTIMODAL REPRESENTATION OF POTENTIAL MOTIFS

We represent residues in a way that captures both structural and sequential context of any potential motif around the residue, so that each residue embedding itself summarizes the local motif.

**Joint encoder.** Let $\mathcal{G}$ be a joint encoder of 3-D structure and 1D sequence (Fig. 2, Point ①):

$$\mathcal{E} = \mathcal{G}(P) = \mathcal{G}(B, \mathbf{S}) \in \mathbb{R}^{L \times d}, \quad (4)$$

where $\mathcal{E} = (\mathcal{E}_1, \ldots, \mathcal{E}_L)$ and $d$ is the embedding dimension.

**Potential-motif representation.** Each vector $\mathcal{E}_i \in \mathbb{R}^d$ contextualizes residue $i \in [L]$ by its local 3-D neighborhood and its placement in the global protein $P$, and is used for both retrieval and emission.

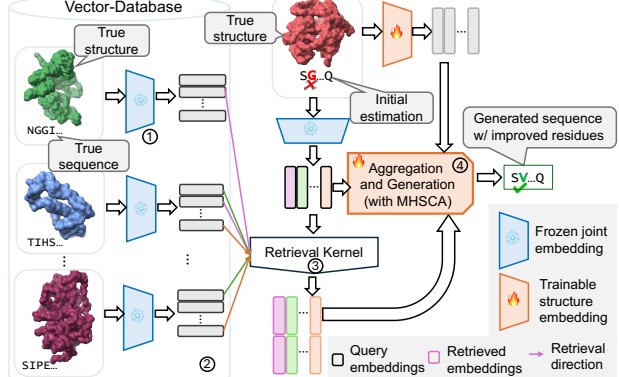

**Query proteins with unknown sequence.** At inference we observe only a query backbone $B^q$. We can sample an initial sequence estimate $\hat{S}^q$ from an off-the-shelf inverse folding model (Sun et al., 2024; Dauparas et al., 2022; Wang et al., 2024b), which we refer to as the *base estimator* in this article. With this, we can form a crude query embedding $\hat{\mathcal{E}}^q = \mathcal{G}(B^q, \hat{S}^q)$, which we treat as a sample from the marginal, i.e., $\hat{\mathcal{E}}^q \sim p(\mathcal{E} \mid B = B^q)$.

Figure 2: The overall pipeline of PRISM. ① We start with a *joint-embedding* model and ② prepare a vector-database by inferring embeddings of known structure–sequence pairs. ③ Our retriever operates on per-token (fine-grained) embeddings, representing the surrounding *potential motifs*. The color coding shows the retrieved vectors for each corresponding site. ④ A hybrid decoder aggregates the retrieved entities and generates a refined protein sequence, enriched with the 3-D structure encoding of the input protein.

### 3.4 Vector Database of Potential Motifs

We treat the vector database $D$ as a *prior-knowledge memory* of potential-motif representations over which retrieval is performed. Given $M$ proteins with structures and sequences $\mathbb{P} = \{(B^p, S^p) : p \in [M]\}$, we encode each $P^p$ via Eq. 4 to obtain $\mathbb{E} = \{\mathcal{E}^p\}_{p=1}^M$. The database is

$$D = \{ d = (\mathcal{E}_r^p, r, p) \ : \ p \in [M], \ r \in [|P^p|] \}.$$

Each residue embedding $\mathcal{E}_r^p$ summarizes the locality around residue $r$ in protein $p$. Let $\phi(\cdot)$ map a residue neighborhood to a motif representation in a metric space $(\mathcal{M}, d)$. Retrieval by similarity of $\mathcal{E}_i$ to $\mathcal{E}(d)$ effectively searches for nearby motifs in $\mathcal{M}$. *Implementation note:* Our vector-DB is created with the *training split of the CATH-4.2 dataset* (Orengo et al., 1997), and any search in it runs entirely on GPU, substantially reducing search time (Sec. 4.2.3).

### 3.5 Retrieval Kernel

The retrieval kernel $p(\mathcal{R} \mid \mathcal{E}, D)$ may be instantiated in multiple ways consistent with the latent-variable formulation. In Appendices D and E.2 we outline representative, non-exhaustive choices for stochastic and deterministic variants.

A *trainable stochastic kernel* constitutes the most faithful realization of the graphical model, defining a proper distribution over residue-level neighbors and enabling learned retrieval priors (Appendix E.2). A *deterministic* $\mathrm{Top}K$ operator offers an efficient approximation (Appendix D, Eq. 20). Formally,

$$p(\mathcal{R} \mid \mathcal{E}, D) = \prod_{i=1}^{L} p(\mathcal{R}_i \mid \mathcal{E}_i, D), \quad p(\mathcal{R}_i \mid \mathcal{E}_i, D) = \delta\big(\mathcal{R}_i - \mathrm{Top}K(\mathcal{E}_i; D)\big), \tag{5}$$

with Dirac distribution $\delta(\cdot)$. Both variants arise from the same underlying model class, differing only in how the latent retrieval variable $R$ is instantiated.

While the trainable variant provides a direct path toward more expressive retrieval mechanism that can potentially lead to improved performance at the expense of additional computing resource (Tab. 9), the deterministic approximation yields an efficient yet powerful solution (Tab. 1–5 in Sec. 4.2).

### 3.6 Attribution Marginal

Retrieval provides candidates but not how they are used. We realize *attribution* via attention weights computed by $T$ hybrid transformer blocks in our aggregation-and-generation module $F_{\theta_\mathbf{Z}}$ (Sec. F), parameterized by $\theta_\mathbf{Z}$. Each head $h \in \{1, \dots, H\}$ in block $t \in \{1, \dots, T\}$ computes $\alpha_{ik}^{(t,h)} = \mathrm{softmax}_k\big(\langle q_i^{(t,h)}, k_{ik}^{(t,h)} \rangle / \sqrt{d_h}\big)$, with query vector $q_i^{(t,h)}$ for residue $i$ and key $k_{ik}^{(t,h)}$ for neighbor $R_{ik}$. Thus $\mathbf{Z}$ is a deterministic function $\mathcal{A}(\mathcal{R}, \mathcal{E}, B)$:

$$\{\alpha_{ik}^{(t,h)}\}_{i,k,t,h} = \mathcal{A}(\mathcal{R}, \mathcal{E}, B), \qquad p(\mathbf{Z} \mid \mathcal{R}, \mathcal{E}, B) = \delta\big(\mathbf{Z} - \mathcal{A}(\mathcal{R}, \mathcal{E}, B)\big). \tag{6}$$

Fig. 7 details the hybrid self-/cross-attention design. In Sec. 4.3.5 we provide ablation to demonstrate the effectiveness of this design.

### 3.7 Sequence Emission

Given $B$ and $\mathcal{R}$, the module $F_{\theta_\mathbf{Z}}$ forms retrieval-aware residue representations through $\mathbf{Z}$ and outputs per-residue logits $\mathbf{Y}(\mathcal{E}, B, \mathcal{R}) = F_{\theta_\mathbf{Z}}\big(F_{\theta_B}(B), \mathcal{E}, \mathcal{R}\big) \in \mathbb{R}^{L \times 20}$, where $F_{\theta_B}$ is a structure encoder (Appendix. F). The emission distribution factorizes:

$$p(\mathbf{S} \mid \mathcal{E}, B, \mathcal{R}, \mathbf{Z}) = \prod_{i=1}^{L} \mathrm{Cat}\big(S_i; \mathrm{softmax}(\mathbf{Y}(\mathcal{E}, B, \mathcal{R}))_i\big). \tag{7}$$

*Remark.* Although we write $p(\mathbf{S} \mid \mathcal{E}, B, \mathcal{R}, \mathbf{Z})$, the logits $\mathbf{Y}(\mathcal{E}, B, \mathcal{R})$ already incorporate the deterministic attribution $\mathbf{Z}$ computed by $F_{\theta_\mathbf{Z}}$.

### 3.8 Training Objective

Under our *deterministic reduction* (Appendix E.7), the objective collapses to maximizing the standard log-likelihood via per-residue cross-entropy, $\hat{\theta} = \arg\max_\theta \mathbb{E}_p\big[\log p_\theta(\mathbf{S} \mid \cdot)\big]$, with learnable

Table 1: Comparison of protein inverse folding methods on the CATH-4.2 test split (Short, Single-chain, and All). Best and second-best scores are shown in **bold** and *italic*. Foldability metrics are computed using ESMFold (Lin et al., 2023), and the subscript $aido$ denotes AIDO.Protein-IF was used as the base estimator.

| Method | Uses pLM | Uses RAG | Short | | Single | | All | | | | |
|--------|----------|----------|-------|-------|--------|-------|-----|-----|------|-------|--------|
| | | | PPL ↓ | AAR % ↑ | PPL ↓ | AAR % ↑ | PPL ↓ | AAR % ↑ | RMSD ↓ | sc-TM ↑ | pLDDT ↑ |
| StructTrans | ✗ | ✗ | 8.39 | 28.14 | 8.83 | 28.46 | 6.63 | 35.82 | - | - | - |
| GVP | ✗ | ✗ | 7.23 | 30.60 | 7.84 | 28.95 | 5.36 | 39.47 | - | - | - |
| ProteinMPNN | ✗ | ✗ | 6.21 | 36.35 | 6.68 | 34.43 | 4.61 | 45.96 | - | - | - |
| ProtMPNN-CMLM | ✗ | ✗ | 7.16 | 35.42 | 7.25 | 35.71 | 5.03 | 48.62 | 1.64 | 0.910 | 0.812 |
| PRISM (str. enc.) | ✗ | ✗ | 4.26 | 35.29 | 3.40 | 48.97 | 3.39 | 49.17 | - | - | - |
| PiFold | ✗ | ✗ | 6.04 | *39.84* | 6.31 | 38.53 | 4.55 | 51.66 | 1.64 | 0.911 | 0.816 |
| LM-Design | ✓ | ✗ | 7.01 | 35.19 | 6.58 | 40.00 | 4.41 | 54.41 | - | - | - |
| DPLM | ✓ | ✗ | - | - | - | - | - | 54.54 | - | - | - |
| AIDO.Protein-IF | ✓ | ✗ | *4.09* | 38.46 | *2.91* | *58.87* | *2.94* | *58.60* | *1.58* | *0.912* | *0.831* |
| PRISM$_{aido}$ | ✓ | ✓ | **3.74** | **40.98** | **2.68** | **60.89** | **2.71** | **60.43** | **1.56** | **0.913** | **0.835** |

parameters $\theta = \{\theta_{\mathbf{Z}}, \theta_B\}$. When the retrieval prior is *trainable* (Appendix E.2), the objective is augmented with a KL-regularization term defined with respect to an amortized posterior $q(\mathcal{R} \mid \cdot)$, i.e., $\hat{\theta} = \arg\max_{\theta} \mathbb{E}_q\big[\log p_{\underline{\theta}}(\mathbf{S} \mid \cdot)\big] - \mathbb{E}_q\big[\mathrm{KL}\big(q(\mathcal{R} \mid \cdot) \,\|\, p_{\underline{\theta}}(\mathcal{R} \mid \cdot)\big)\big]$, with $\underline{\theta} = \{\theta_{\mathbf{Z}}, \theta_B, \theta_{\mathcal{G}}\}$, where $\theta_{\mathcal{G}}$ represents the learnable parameters of the joint encoder $\mathcal{G}$. We leave the exploration of jointly learning amortized posteriors for $\mathcal{E}$ and $\mathbf{Z}$ via their KL terms to future work.

# 4 EXPERIMENTS AND RESULTS

## 4.1 EXPERIMENTAL SETUP

**Datasets.** We evaluate PRISM on several widely used benchmarks: CATH-4.2, TS50, TS500, CAMEO 2022, PDB date split, and orphan proteins dataset. We use the CATH-4.2 training split both for training and to create the vector database, preventing any form of data leakage *by construction* (Orengo et al., 1997). TS50 and TS500 are used only for evaluation to test cross-dataset generalization (Li et al., 2014), while CAMEO 2022 and the PDB date split assess robustness on proteins outside the CATH classification and under temporally disjoint conditions (Campbell et al., 2024). To further test PRISM's generalizability to entirely novel backbones, we evaluate it on the orphan proteins dataset by Jing et al. (2024). Full dataset descriptions and statistics are provided in Appendix B.

Table 2: Comparison on TS50 and TS500.

| Models | TS50 | | TS500 | |
|--------|------|------|-------|------|
| | PPL ↓ | AAR % ↑ | PPL ↓ | AAR % ↑ |
| GVP | 4.71 | 44.14 | 4.20 | 49.14 |
| ProteinMPNN | 3.93 | 54.43 | 3.53 | 58.08 |
| ProtMPNN-CMLM | 3.46 | 53.68 | 3.35 | 56.45 |
| PRISM (str. enc.) | 3.10 | 54.41 | 2.88 | 57.66 |
| PiFold | 3.86 | 58.72 | 3.44 | 60.42 |
| LM-Design | 3.82 | 56.92 | **2.13** | 64.50 |
| AIDO.Protein-IF | *2.68* | *66.19* | *2.42* | *69.66* |
| PRISM$_{aido}$ | **2.43** | **67.92** | *2.27* | **70.53** |

**Evaluation Metrics.** We report two sequence-level metrics and three structure-level metrics. Sequence accuracy is assessed by *amino acid recovery (AAR)* and *perplexity (PPL)*, while foldability is assessed with *RMSD*, *sc-TM*, and *pLDDT*. Together these capture both sequence correctness and structural realizability. Detailed formulations of all metrics are provided in Appendix B.

Table 3: Comparison on CAMEO 2022 and PDB date split. Multiflow, ESM3, and DPLM-2 results are taken from (Wang et al., 2024a).

| Models | CAMEO 2022 | | PDB date split | |
|--------|------------|------|----------------|------|
| | PPL ↓ | AAR % ↑ | PPL ↓ | AAR % ↑ |
| ProtMPNN-CMLM | 3.62 | 50.14 | 3.42 | 52.98 |
| PRISM (str. enc.) | 3.20 | 51.20 | 3.04 | 53.85 |
| MultiFlow | – | 33.58 | – | 37.59 |
| ESM3 | – | 46.24 | – | 49.42 |
| DPLM2-3B | – | *53.73* | – | *57.91* |
| AIDO.Protein-IF | *2.68* | 63.52 | *2.49* | 66.27 |
| PRISM$_{aido}$ | **2.53** | **64.63** | **2.35** | **67.47** |

**Baselines.** We compare against a comprehensive suite of state-of-the-art inverse folding methods, including StructTrans, GVP, ProteinMPNN, ProteinMPNN-CMLM, PiFold, LM-Design, DPLM, MultiFlow, ESM-3, DPLM-2, and the large-scale AIDO.Protein, which we also adopt as our *joint encoder*. Appendix C provides details of these baselines.

## 4.2 RESULTS AND DISCUSSION

### 4.2.1 RESULTS ON CATH 4.2, TS50, TS500, CAMEO 2022, AND PDB DATE SPLIT

**CATH-4.2** (Tab. 1): shows that PRISM consistently improves over strong baselines across all three CATH-4.2 settings. Compared to AIDO.Protein-IF, PRISM reduces perplexity from 4.09 to 3.74 on short-chains, from 2.91 to 2.68 on single-chains, and from 2.94 to 2.71 on the full test set. These PPL gains also translate to higher recovery. Specifically, AAR increases by 2.52 (40.98 vs. 38.46) on short, 2.02 (60.89 vs. 58.87) on single-chain, and 1.83 (60.43 vs. 58.60) on all. Notably, even PRISM

Table 4: Foldability comparison using AF2 protein folding model. The median and the mean are provided outside and inside the parenthesis, respectively.

| Models | CAMEO 2022 | | | PDB date split | | |
|---|---|---|---|---|---|---|
| | RMSD ↓ | sc-TM ↑ | pLDDT ↑ | RMSD ↓ | sc-TM ↑ | pLDDT ↑ |
| DPLM2-3B | 1.67 (1.833) | 0.926 (0.846) | 0.923 (0.898) | 1.21 (1.399) | 0.954 (0.918) | 0.944 (0.919) |
| AIDO.Protein-IF | 1.54 (1.665) | 0.942 (0.862) | 0.932 (**0.916**) | 1.1 (1.231) | 0.963 (0.936) | **0.953** (0.937) |
| PRISM$_{aido}$ | **1.49 (1.621)** | **0.948 (0.867)** | **0.934** (0.916) | **1.04 (1.2)** | **0.964 (0.938)** | 0.953 (0.938) |

Table 5: Runtime analysis (in seconds per protein) across different benchmarks. We decompose runtime into the base estimator (AIDO.Protein-IF), retrieval, and decoding. The total time is the sum of all components.

| Model | TS50 | TS500 | CAMEO2022 | PDB date split | CATH 4.2 test | CATH 4.2 val | Average |
|---|---|---|---|---|---|---|---|
| Base estimator (AIDO.Protein-IF) | 0.83 | 1.03 | 0.99 | 0.91 | 0.87 | 0.89 | 0.92 |
| +Retrieval | $3.1e^{-3}$ | $1.1e^{-3}$ | $1.3e^{-3}$ | $6.0e^{-4}$ | $5.0e^{-4}$ | $6.0e^{-4}$ | $1.2e^{-3}$ |
| +Decoding | 0.08 | 0.17 | 0.17 | 0.12 | 0.10 | 0.11 | 0.13 |
| Total | 0.91 | 1.20 | 1.17 | 1.03 | 0.97 | 1.00 | 1.05 |

(str. enc.), the structure-only variant, already outperforms its corresponding baseline, ProteinMPNN-CMLM. while our full framework, with AIDO.Protein-IF as the base estimator and joint encoder, yields the best overall trade-off in sequence-level and structure-level metrics. All scores are obtained with deterministic decoding, where we use the deterministic approximation of our retriever and chose `argmax` sampling with the final logits. Additional results on this dataset, including threshold-based designability analysis, are provided in Appendix L and Tab. 17.

**TS50 and TS500** (Tab. 2): On TS50, PRISM sets new SoTA on both metrics, with PPL 2.43 vs. 2.68, and AAR of 67.92 vs. 66.19 consistently improving over its base estimator AIDO.Protein-IF. On TS500, PRISM achieves the best AAR (70.53) and a strong PPL of 2.27, while LM-Design reports a lower PPL on TS500, its AAR is substantially lower (64.50), indicating that PRISM's conditioning yields sequences that align better with native residues.

**CAMEO 2022 and PDB date split** (Tab. 3): PRISM improves both confidence and recovery on distribution shifts. On CAMEO 2022, PPL decreases from *2.68* (AIDO.Protein-IF) to 2.53, while AAR rises improves by 1.11. On the PDB date split, PPL drops from 2.49 to 2.35 and AAR improves from 66.27 to 67.47. These trends on these four stand-alone test sets underscore that our proposed approach contributes stable gains even when test distributions diverge from CATH-4.2.

### 4.2.2 FOLDABILITY ANALYSIS

Tab. 4 and Appendix Tab. 14 show end-to-end foldability of designed sequences via AlphaFold2 (John et al., 2021). PRISM consistently improves structural fidelity over AIDO.Protein-IF across datasets: on TS50, RMSD drops from 1.075 to 0.985, sc-TM rises from 0.956 to 0.964, and pLDDT slightly improves (0.949→0.950); on TS500, RMSD improves (1.18→1.125) with sc-TM also higher (0.964). On CAMEO 2022 and the PDB date split, PRISM attains the best RMSD and sc-TM alongside competitive or best pLDDT. These consistent gains indicate that PRISM's higher AAR is not merely superficial residue matching, rather it translates to sequences that fold closer to the target backbones with stronger global topology (sc-TM) and comparable or better local accuracy (pLDDT).

### 4.2.3 RUNTIME ANALYSIS

A key advantage of PRISM is that its substantial accuracy gains come at negligible runtime cost. As shown in Tab. 5, the base estimator (AIDO.Protein-IF) requires on average 0.92 seconds per protein, while our full framework adds only lightweight

Table 6: Inverse folding performance on the orphan proteins dataset (Jing et al., 2024). Here ESMFold (Lin et al., 2023) was used as the protein folding model.

| Model | AAR (%) ↑ | sc-TM ↑ | RMSD ↓ | pLDDT ↑ |
|---|---|---|---|---|
| Native Sequence | 100.00 | 0.360 (0.449) | 3.61 (3.654) | 0.457 (0.535) |
| AIDO.Protein-IF | 28.05 (30.22) | 0.369 (0.420) | 3.78 (3.615) | 0.538 (0.535) |
| PRISM$_{aido}$ | 28.33 (30.85) | 0.391 (0.437) | 3.92 (3.846) | 0.556 (0.531) |
| ProteinMPNN-CMLM | 29.27 (31.24) | 0.359 (0.428) | 3.60 (3.712) | 0.569 (0.525) |
| PRISM$_{mpnn}$ | 27.12 (30.76) | 0.404 (0.437) | 3.60 (3.587) | 0.533 (0.538) |

retrieval ($\sim 1.2 \times 10^{-3}$s) and decoding (0.13s), resulting in a total runtime of 1.05s. This corresponds to a relative overhead of merely 14.3% compared to the base estimator.

In contrast, the improvements in accuracy are much larger. Averaged across benchmarks (Appendix Tab. 18), PRISM reduces perplexity from 2.68 to 2.43 (9.3% improvement) and boosts AAR from 63.0% to 66.9% (+3.9 absolute points). In other words, PRISM delivers significant and consistent accuracy gains across all test sets while incurring only a negligible runtime overhead. This balance demonstrates the efficiency of memory-based retrieval: it enriches the model's representations without

sacrificing throughput, making PRISM a practically viable and scientifically impactful extension over the base estimator.

### 4.2.4 ORPHAN PROTEIN DESIGN

Because natural-sequence recovery does not fully capture generalization to novel backbones, we additionally evaluate PRISM on the orphan proteins dataset Jing et al. (2024), a set of eleven proteins with no detectable sequence homologs in major databases. As shown in Tab. 6, PRISM consistently improves over its respective base estimators and achieves foldability scores close to the native sequences, despite the lack of homologous entries in its retrieval database. This indicates that PRISM is not merely memorizing natural sequences, but can leverage retrieved structural context to enhance design even for highly novel backbones.

### 4.3 ABLATION STUDIES AND ADDITIONAL ANALYSES

To better understand the contributions of individual components and design choices in PRISM, we conduct a series of ablation studies and supplementary analyses. These highlight the effectiveness, efficiency, and robustness of our framework.

### 4.3.1 ABLATION THE NUMBER OF RETRIEVED ENTRIES

We conducted an ablation study to analyze the effect of the number of retrieved vectors $K$ on model performance. As shown in Fig. 3, increasing $K$ consistently reduces perplexity (PPL) on the CATH-4.2 validation split. The improvement is sharp for small $K$ (e.g., from 2.788 at $K = 1$ to 2.709 at $K = 5$), but gradually saturates as $K$ increases further. Beyond $K \geq 35$, PPL stabilizes around 2.681, showing no further significant gains. Therefore, we choose $K = 35$ as the optimal setting, striking a balance between efficiency and accuracy.

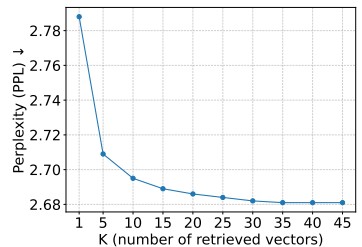

Figure 3: Ablation on $K$ (CATH-4.2 validation split).

### 4.3.2 EFFECT OF PROTEIN SIZE ON RECOVERY

Fig. 4 shows the distribution of amino-acid recovery rates (AAR) across protein length bins on the CATH-4.2 test set. PRISM consistently outperforms AIDO.Protein-IF across all lengths, with particularly notable gains for shorter proteins ($< 200$ residues) where inverse folding is more challenging. These results confirm that PRISM's improvements are robust across varying sequence lengths, rather than confined to a narrow subset of proteins.

### 4.3.3 CONTRIBUTION OF RETRIEVAL

A central question in our study is whether retrieval itself contributes meaningfully to inverse folding, beyond what large pretrained models or structural encoders already achieve. Across all benchmarks, PRISM with retrieval consistently outperforms AIDO.Protein-IF in both perplexity and recovery metrics (Tab. 1, 2, 3, 4). For instance, on CATH-4.2 PRISM improves

Figure 4: AAR distribution across protein length bins on CATH-4.2. PRISM consistently outperforms AIDO.Protein, with especially large gains for shorter proteins ($<$ 200 residues).

AAR by nearly two percentage points over AIDO.Protein-IF, while on TS50 and TS500 it reduces perplexity and boosts recovery simultaneously–demonstrating that retrieval provides tangible benefits across datasets of varying scale and diversity.

To isolate the effect of retrieval from that of multimodal representation, we introduce a controlled *unimodal* ablation variant, PRISM (str. enc.), which replaces the AIDO.Protein-IF joint encoder with a purely structure-based encoder (ProteinMPNN-CMLM) and performs retrieval solely over structural embeddings. Remarkably, even in this restricted setting, our retrieval mechanism delivers consistent gains over the baseline ProteinMPNN-CMLM across all datasets (Tab. 1, 2, 3). This result isolates retrieval as an independent driver of performance – even without sequence-level priors, fine-grained retrieval improves recovery by supplying complementary local context that a single encoder cannot capture. Together, these findings establish retrieval to not be an auxiliary feature, but as a core contributor in PRISM.

Table 7: Ablation on hybrid-attention vs cross-attention-only.

| Models | TS50 | | TS500 | | CATH-4.2 test split | | CATH-4.2 val split | |
|---|---|---|---|---|---|---|---|---|
| | PPL ↓ | AAR % ↑ | PPL ↓ | AAR % ↑ | PPL ↓ | AAR % ↑ | PPL ↓ | AAR % ↑ |
| PRISM (w/o MHSA) | 2.56 | 64.23 (65.12/64.98) | 2.36 | 69.94 (68.43/70.04) | 2.82 | 59.26 (57.44/60.41) | 2.79 | 59.51 (58.42/61.11) |
| PRISM (full) | **2.43** | **67.92** (66.98/66.70) | **2.27** | **70.53** (69.57/70.97) | **2.71** | **60.43** (58.55/61.41) | **2.68** | **60.26** (59.28/61.89) |

Table 8: Ablation on the number of MHSCA blocks.

| # of blocks | TS50 | | TS500 | | CATH-4.2 test split | | CATH-4.2 val split | |
|---|---|---|---|---|---|---|---|---|
| | PPL ↓ | AAR % ↑ | PPL ↓ | AAR % ↑ | PPL ↓ | AAR % ↑ | PPL ↓ | AAR % ↑ |
| N/A (base est.) | 2.68 | 66.19 (64.69/64.66) | 2.42 | 69.66 (68.04/69.60) | *2.94* | *58.60 (57.27/60.13)* | 2.90 | 58.73 (58.00/60.62) |
| 1 | 2.44 | 66.90 (66.84/66.58) | **2.26** | **70.93** (69.58/70.97) | 2.72 | 60.23 (58.53/61.39) | 2.69 | 60.17 (59.20/61.86) |
| 2 | **2.43** | **67.92** (66.98/66.70) | 2.27 | 70.53 (69.57/70.97) | **2.71** | **60.43** (58.55/61.41) | **2.68** | **60.26** (59.28/61.89) |
| 3 | 2.44 | 67.71 (66.75/66.48) | 2.26 | 70.59 (69.58/70.99) | 2.71 | 60.35 (58.59/61.42) | 2.68 | 60.24 (59.30/61.91) |

### 4.3.4 EFFECT OF EXTENDING RETRIEVAL DATABASE

A natural question is whether enlarging the retrieval memory at inference time further improves performance. Our theoretical analysis (Appendix G) establishes that once the vector database achieves near-complete $\varepsilon$-coverage of the motif space, additional entries predominantly duplicate existing motifs and thus provide diminishing returns. Empirically, we confirm this saturation effect: augmenting the database with new PDB entries yields almost identical results across all benchmarks (Appendix G, Tab. 13), with differences well within retrieval noise. For instance, for CAMEO 2022 the AAR remains ~64.6% whether using only the CATH-4.2 memory, the PDB extension, or their combination. This finding highlights that PRISM's fixed vector database already captures the relevant structural landscape, making post-hoc memory growth unnecessary. Crucially, it validates our design choice of treating the vector database as a *prior knowledge store* rather than an ever-expanding index, achieving state-of-the-art recovery while avoiding uncontrolled growth in memory size.

### 4.3.5 CONTRIBUTION OF HYBRID DECODER WITH MHSCA

We next ablate the design of the aggregation module by comparing our hybrid multihead self–cross attention (MHSCA) decoder with a simplified variant that relies only on multihead cross-attention (MHCA). As shown in Tab. 7 and Appendix Tab. 15, removing the self-attention component degrades performance across all benchmarks. While the cross-attention–only variant already improves over the base estimator by attending to retrieved vectors, it lacks the ability to contextualize and refine these fragments jointly. Incorporating MHSA within the block allows the model to propagate information among retrieved neighbors before aligning them with the query, yielding consistent gains. For example, on TS50, the AAR increases from 64.2% to 67.9%, and perplexity drops from 2.56 to 2.43; on the CATH-4.2 test split, AAR rises from 59.3% to 60.4% with a corresponding reduction in PPL (2.82 → 2.71). Similar improvements are observed on TS500 and the PDB date split, with relative gains of +0.8%–1.2% AAR. Importantly, these gains are consistent across both in-distribution (CATH-4.2) and out-of-distribution (CAMEO 2022, PDB date split) settings, highlighting that the hybrid MHSCA architecture provides more expressive aggregation by jointly leveraging self- and cross-attention. This validates the adoption of MHSCA as the default decoding module in PRISM.

Table 9: Effect of finetuning the joint encoder $\mathcal{G}$ (denoted "ft. $\mathcal{G}$") *and a simple inference-time iterative refinement approach* (denoted "iter."). Here "orig. $\mathcal{G}$" denotes the original joint encoder without finetuning.

| Model | PPL ↓ | AAR (%) ↑ |
|---|---|---|
| PiFold | 4.55 | 51.66 |
| PRISM$_{pifold}$ (orig. $\mathcal{G}$) | 2.72 | 60.09 |
| PRISM$_{pifold}$ (orig. $\mathcal{G}$, iter.) | 2.73 | 61.51 |
| PRISM$_{pifold}$ (ft. $\mathcal{G}$) | 2.63 | 62.01 |
| PRISM$_{pifold}$ (ft. $\mathcal{G}$, iter.) | 2.67 | 63.09 |
| AIDO.Protein-IF | 2.94 | 58.60 |
| PRISM$_{aido}$ (orig. $\mathcal{G}$) | 2.71 | 60.43 |
| PRISM$_{aido}$ (orig. $\mathcal{G}$, iter.) | 2.73 | 61.87 |
| PRISM$_{aido}$ (ft. $\mathcal{G}$) | 2.59 | 62.07 |
| PRISM$_{aido}$ (ft. $\mathcal{G}$, iter.) | 2.65 | 63.33 |

### 4.3.6 EFFECT OF AGGREGATION DEPTH (MHSCA LAYERS)

We next study how the number of multihead self–cross attention (MHSCA) blocks in the aggregation module affects performance (Tab. 8 and Appendix Tab. 16). Adding even a single block over the base

Table 10: Ablation on the base estimator. Here the subscripts $pmpnn$, $pifold$, and $aido$ respectively denote whether ProteinMPNN-CMLM, PiFold, or AIDO.Protein-IF were used as the base estimator for PRISM. All results of PRISM here use the original (non-finetuned) joint encoder $\mathcal{G}$.

| Model | CATH 4.2 PPL ↓ | CATH 4.2 AAR (%) ↑ | TS50 PPL ↓ | TS50 AAR (%) ↑ | TS500 PPL ↓ | TS500 AAR (%) ↑ |
|---|---|---|---|---|---|---|
| ProteinMPNN-CMLM (base est.) | 5.03 | 48.62 | 3.46 | 53.68 | 3.35 | 56.45 |
| PRISM$_{pmpnn}$ | **2.82** | **58.02** | **2.52** | **65.03** | **2.35** | **69.0** |
| PiFold (base est.) | 4.55 | 51.66 | 3.86 | 58.72 | 3.44 | 60.42 |
| PRISM$_{pifold}$ | **2.72** | **60.09** | **2.44** | **66.56** | **2.26** | **71.16** |
| AIDO.Protein-IF (base est.) | 2.94 | 58.60 | 2.68 | 66.19 | 2.42 | 69.66 |
| PRISM$_{aido}$ | **2.71** | **60.43** | **2.43** | **67.92** | **2.27** | **70.53** |

Table 11: PRISM's AAR improvements across proteins grouped by PiFold initial prediction quality. Here lower bins correspond to lower-quality initial sequences.

| PiFold AAR Bin (%) | # Proteins | PiFold Mean AAR (%) | $\text{PRISM}_{pifold}$ (original $\mathcal{G}$) Mean AAR (%) | $\Delta$ (original $\mathcal{G}$) | $\text{PRISM}_{pifold}$ (ft. $\mathcal{G}$, iter.) Mean AAR (%) | $\Delta$ (ft. $\mathcal{G}$, iter.) |
|---|---|---|---|---|---|---|
| 0–20 | 12 | 15.89 | 18.48 | 2.59 | 25.93 | 10.03 |
| 20–30 | 74 | 25.97 | 29.52 | 3.55 | 33.79 | 7.82 |
| 30–40 | 131 | 35.51 | 42.63 | 7.12 | 48.44 | 12.94 |
| 40–50 | 272 | 45.95 | 55.22 | 9.27 | 59.21 | 13.26 |
| 50–60 | 464 | 54.88 | 64.60 | 9.72 | 67.53 | 12.66 |
| 60–70 | 157 | 62.79 | 71.47 | 8.68 | 73.42 | 10.63 |
| 70–80 | 10 | 73.57 | 80.03 | 6.46 | 78.76 | 5.18 |

estimator (AIDO.Protein-IF) yields a large gain: on the CATH-4.2 test split, AAR improves from $58.6\%$ to over $60.2\%$, and perplexity drops from 2.94 to 2.72. Increasing to two blocks provides the best overall trade-off, achieving the strongest or tied-best results across nearly all benchmarks (e.g., CAMEO 2022 with PPL 2.53 and AAR $64.6\%$, CATH-4.2 validation with PPL 2.68 and AAR $60.3\%$). Using three blocks maintains similar accuracy but shows no consistent benefit, with small oscillations likely due to noise. These results indicate that the aggregation mechanism quickly saturates, and two MHSCA layers suffice to capture the additional context from retrieved fragments while avoiding redundancy or overfitting.

### 4.3.7 EFFECT OF FINETUNING JOINT ENCODER AND ITERATIVE REFINEMENT

Our latent-variable formulation naturally allows finetuning the joint encoder $\mathcal{G}$ via a trainable retrieval kernel (Appendix E.2). To assess its impact, we evaluate PRISM with and without finetuning $\mathcal{G}$. As shown in Table 9, finetuning $\mathcal{G}$ yields consistent improvements across two base estimators of different strengths: PiFold and AIDO.Protein-IF, respectively denoted with subscripts $pifold$ and $aido$. This result indicates that adapting the joint encoder further enhances the retrieval, and subsequently, the generation quality.

We additionally ablate a simple inference-time refinement procedure, inspired by recent approaches (Zheng et al., 2023; Wang et al., 2024a). Here, the output sequence is iteratively fed back into the PRISM framework to improve the generation. This refinement tends to further increase AAR, with only a marginal change in PPL, suggesting that PRISM can effectively leverage its own predictions to refine sequences without any extra training. A more systematic study of different refinement paradigms is left to future work.

### 4.3.8 IMPACT OF INITIAL SEQUENCE QUALITY

To investigate the sensitivity of PRISM's final performance to the quality of initial sequence estimate, we first compare PRISM across three different base estimators of varying strengths: ProteinMPNN-CMLM, PiFold, and AIDO.Protein-IF. The results, summarized in Tab. 10, show that PRISM improves each of these estimators by a large margin. Remarkably, $\text{PRISM}_{pifold}$ and $\text{PRISM}_{pmpnn}$ (both using single-iteration refinement) already achieve performance comparable to the stronger AIDO.Protein-IF baseline, which itself uses iterative refinement. These results clearly indicates that PRISM effectiveness is not tied to AIDO.Protein-IF or any specific initializer. Because PRISM is base estimator agnostic, it can in principle be paired with surface-based generators such as SurfPro (Song et al., 2024), SurfDesign (Wu et al., 2024), or biochemistry-aware generators such as BC-Design (Tang et al., 2025). Such combinations would allow PRISM to retrieve surface- or chemistry-aligned exemplars, extending its benefits beyond backbone-conditioned inverse folding.

To further assess PRISM's robustness to, the quality of the initial sequence provided by the base estimator, we stratify proteins by their PiFold AAR and evaluate PRISM's performance within each bin. This analysis reveals how much improvement PRISM achieves when starting from weak, moderate, or strong initial estimates. As shown in Table 11, PRISM consistently boosts AAR across all bins, with particularly large gains in the low-AAR regimes.

## 5 CONCLUSIONS

We present PRISM, a multimodal retrieval-augmented framework for protein inverse folding that integrates fine-grained retrieval of potential motif embeddings with a hybrid self-cross attention decoder. PRISM achieves new state of the art across multiple benchmarks in sequence recovery and foldability, while adding only negligible runtime overhead. Our latent-variable formulation provides theoretical grounding, and ablations confirm the central role of different design choices, including retrieval, hybrid attention, and aggregation mechanism. These results establish fine-grained retrieval as a principled and scalable approach for advancing protein sequence design.

## ACKNOWLEDGMENTS

We thank the anonymous reviewers for their valuable insights and recommendations, which have greatly improved our work. This research has been graciously funded by the National Science Foundation (NSF) awards CNS2414087 and IIS2123952; by the Defense Advanced Research Projects Agency (DARPA) award HR00112390063; and, the Semiconductor Research Corporation (SRC) AIHW award 2024AH3210. Any opinions, findings, and conclusions or recommendations expressed in this publication are those of the author(s) and do not necessarily reflect the views of the National Science Foundation, the Defense Advanced Research Projects Agency, and the Semiconductor Research Corporation.

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

## A    RELATED WORKS

Protein inverse folding, the process of designing amino acid sequences that fold into specific three-dimensional structures, has been a focal point of computational biology research (Gao et al., 2022; Mahbub et al., 2025; Sun et al., 2024; Hsu et al., 2022; Zheng et al., 2023). In 2022, Dauparas et al. (2022) proposed ProteinMPNN, widely popular autoregressive method for designing protein sequences that fold into desired structures. It achieved an impressive sequence recovery rate on native backbones, outperforming traditional methods, showing versatility extending to designing monomers, cyclic oligomers, nanoparticles, and target-binding proteins. Gao et al. (2022) introduced PiFold, a method that effectively combines expressive features with an autoregressive sequence decoder to enhance both the accuracy and efficiency of protein design. PiFold achieved a high recovery rate on the benchmark dataset and demonstrated a speed advantage, being 70 times faster than some autoregressive counterparts. That same year, Hsu et al. (2022) proposed a sequence-to-sequence transformer model trained using predictions by AlphaFold2, a state-of-the-art structure prediction method (John et al., 2021). By leveraging putative structures of millions of proteins, their approach achieved a notable improvement in the field. Zheng et al. (2023) introduced the usage of protein language models (Nadav et al., 2023; Meier et al., 2021) for structure-conditioned protein sequence design, or in other words, inverse folding. Another work by Wang et al. (2024a) extended this by incorporating diffusion language modeling for effective sequence generation. Sun et al. (2024) pretrained a 16 billion parameter protein language model with a mixture-of-expert architecture, which they further adapted for prediction and sequence generation tasks, and surpassing the previous methods. To address the need for standardized evaluation, Gao et al. (2023) also proposed ProteinInvBench, a comprehensive benchmark for protein design. This framework includes extended design tasks, integrated models, and diverse evaluation metrics, facilitating more rigorous comparisons across different methods. Mao et al. (2023) proposed VFN-IF and and its ESM-equipped variant VFN-IFE, advancing inverse folding by using Vector Field Networks to perform learnable coordinate-based vector operations that jointly model residue frames and atomic geometry. Recent work has also explored *surface-based* and *biochemistry-aware* protein design, such as SurfPro (Song et al., 2024) and SurfDesign (Wu et al., 2024), which operate directly on molecular surfaces, and BC-Design (Tang et al., 2025), which incorporates explicit biochemical constraints into the generation process. These approaches differ from inverse folding models in their use of geometric or physicochemical signals rather than backbone-sequence likelihood, and represent an orthogonal line of structure-conditioned design.

## B    EXPERIMENTAL SETUP

### B.1    DATASETS

We evaluate our framework on six widely used benchmarks: CATH-4.2 (Orengo et al., 1997), TS50 (Li et al., 2014), TS500 (Li et al., 2014), CAMEO 2022 (Campbell et al., 2024), PDB date split (Campbell et al., 2024), and orphan proteins dataset (Jing et al., 2024).

CATH-4.2 is a standard benchmark containing proteins with fewer than 500 residues, and is widely adopted for training, validation, and testing of inverse folding models (Zheng et al., 2023; Wang et al., 2024a). Following prior work, we further analyze three subsets of the CATH-4.2 test set: *short chains* (length < 100, ∼16.5%), *single chains* (∼92.86%), and the full test split. Appendix Fig. 5 shows the sequence length distribution.

TS50 is a compact benchmark of 50 proteins (maximum length 173), while TS500 provides greater variability, ranging from very short chains (43 residues) to long proteins (>1600 residues). Following

Table 12: Statistics of CATH-4.2, TS50, TS500, CAMEO 2022, PDB date split, and Orphan proteins benchmark datasets. Here "seq.", "res.", "len.", and "St. Dev." represent "sequence", "residue", "length", and "standard deviation", respectively.

| Data split | # of seq. | # of res. | Mean Len. | Median Len. | St. Dev. Len. |
|---|---|---|---|---|---|
| CATH-4.2 Train | 18,024 | 3,941,775 | 218.7 | 204.0 | 109.93 |
| CATH-4.2 Validation | 608 | 105,926 | 174.22 | 146.0 | 92.44 |
| CATH-4.2 Test | 1,120 | 181,693 | 162.23 | 138.0 | 82.22 |
| CATH-4.2 Combined | 19,752 | 4,229,394 | 214.12 | 196.0 | 109.06 |
| TS50 | 50 | 6,861 | 137.22 | 145.0 | 25.96 |
| TS500 | 500 | 130,960 | 261.92 | 225.0 | 167.30 |
| CAMEO 2022 | 183 | 44,539 | 243.38 | 228.0 | 144.86 |
| PDB date split | 449 | 86,698 | 193.09 | 178.0 | 81.06 |
| Orphan proteins | 11 | 1,310 | 119.09 | 124.0 | 55.04 |

convention (Zheng et al., 2023; Gao et al., 2022), we use these only for evaluation after training on the CATH-4.2 training split, thereby testing cross-dataset generalization.

CAMEO 2022 comprises 183 recently released structures (average length 243 residues), providing an evaluation on proteins outside the CATH classification and closer to real-world modeling targets (Campbell et al., 2024). The PDB date split (449 proteins; mean length 193) follows the protocol of previous studies such as Campbell et al. (2024) and Wang et al. (2024b), where training and evaluation proteins are separated strictly by deposition date in the Protein Data Bank. This ensures robustness against temporal leakage and simulates forward-looking generalization.

The recovery rate of natural proteins does not fully assess the ability of inverse folding models to generalize to novel or orphan backbones. We therefore evaluate PRISM on the *orphan proteins* dataset prepared by Jing et al. (2024), which contains 11 challenging proteins with no detectable sequence homologs in major databases. Since our retrieval index is built from the CATH-4.2 training split (PDB-derived), these orphan proteins are *guaranteed to be non-overlapping* with both our training set and retrieval index.

## B.2 Evaluation Metrics

We report two sequence-level metrics and three structure-level metrics.

**Sequence-level metrics.** *Amino Acid Recovery (AAR)*: Median sequence recovery is the most widely used metric for inverse folding (Zheng et al., 2023; Wang et al., 2024a; Sun et al., 2024). It measures the percentage of positions where the predicted amino acid matches the native sequence:

$$\text{AAR} = \text{median}\left(\frac{1}{L}\sum_{i=1}^{L}\mathbf{1}(\hat{S}_i = S_i) \times 100\%\right), \tag{8}$$

where $L$ is the protein length and $\mathbf{1}$ is the indicator function.

*Perplexity (PPL)*: Perplexity evaluates how confidently a model predicts the native sequence (Wang et al., 2024b; Tian et al., 2026; Zou et al., 2024). For autoregressive models:

$$\text{PPL}_{AR} = \exp\left(-\frac{1}{\sum_{j=1}^{M}L_j}\sum_{j=1}^{M}\sum_{i=1}^{L_j}\log P(S_i \mid S_{<i}, B)\right). \tag{9}$$

For our non-autoregressive setting:

$$\text{PPL}_{NAR} = \exp\left(-\frac{1}{\sum_{j=1}^{M}L_j}\sum_{j=1}^{M}\sum_{i=1}^{L_j}\log P(S_i \mid \hat{S}, B)\right), \tag{10}$$

where $\hat{S}$ is a noisy initialization of the native sequence.

**Structure-level metrics.** To evaluate whether generated sequences are *foldable* into the target backbone, we use three complementary metrics, following recent studies (Dauparas et al., 2022; Zheng et al., 2023; Wang et al., 2024a;b):

*Root-Mean-Square Deviation (RMSD):* measures the average distance (in Å) between backbone alpha carbon atoms of the predicted and native structures after optimal alignment. Lower RMSD indicates higher structural fidelity.

$$\text{RMSD}(X, Y) = \sqrt{\frac{1}{N} \sum_{i=1}^{N} d_i^2}, \tag{11}$$

where $d_i$ is the distance between the alpha carbons of $i$-th pair of residues in two structures $X$ and $Y$.

*TM-score (sc-TM):* a length-normalized similarity metric that is robust to protein size and is widely used to assess global fold correctness; values $> 0.5$ typically indicate correct fold topology.

$$\text{sc-TM}(X, Y) = \max_{\text{alignments}} \frac{1}{L_{\text{target}}} \sum_{i=1}^{L_{\text{aligned}}} \frac{1}{1 + \left( \frac{d_i}{d_0(L_{\text{target}})} \right)^2} \tag{12}$$

where $d_0(L) = 1.24(L - 15)^{1/3} - 1.8$

*Predicted Local Distance Difference Test (pLDDT):* a per-residue confidence score from AlphaFold2 (John et al., 2021), used here to assess the stability and reliability of folded structures generated from designed sequences.

$$\text{pLDDT} = \frac{1}{L} \sum_{i=1}^{L} \text{conf}(i) \tag{13}$$

Together, these metrics evaluate both *sequence accuracy* and *structural realizability*, which is critical in physical sciences applications of protein design.

### B.3 HARDWARE SPECIFICATIONS

These experiments were conducted with single AMD Milan processor (64-core 2.55 GHz), 256 GB of RAM, and four NVIDIA A100 GPUs (each with 40 GB VRAM). The vector-database takes about 15.1 GB of hard disk space.

### B.4 IMPLEMENTATION DETAILS OF VECTOR DATABASE

Our vector database is created with the CATH-4.2 training split, where each residue corresponds to a database entry as defined in Sect. 3.4 (a total of 3,941,775 residues). Since we leverage this dataset to train our model as well, we additionally mask the query sequence itself to prevent trivial self-retrieval during training.

## C BASELINES

StructTrans (Ingraham et al., 2019) proposed a conditional generative model for protein sequences given 3D structures based on graph representations. GVP (Jing et al., 2020) introduced geometric vector perceptrons, which extend standard dense layers to operate on collections of Euclidean vectors. ProteinMPNN (Dauparas et al., 2022) proposes an autoregressive protein sequence generation approach conditioned on structure. ProteinMPNN-CMLM (Zheng et al., 2023), a non-autoregressive variant of the original ProteinMPNN, has been trained with the conditional masked language modeling (CMLM) objective (Ghazvininejad et al., 2019) and achieves higher score than the original version. LM-Design (Zheng et al., 2023) is another non-autoregressive model trained with CMLM that leverages pretrained protein language models for inverse folding. DPLM (Wang et al., 2024a) extends this work by using discrete diffusion language modeling objective to enhance sequence generation capabilities of languange models. Multiflow (Campbell et al., 2024), ESM3 (Hayes et al., 2025),

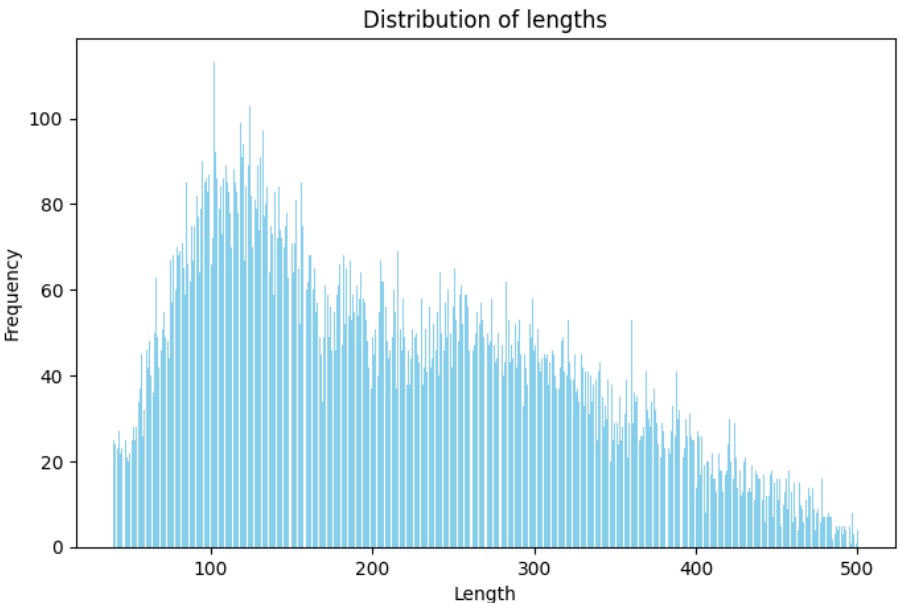

Figure 5: Distribution of lengths of the protein sequences in the benchmark dataset CATH-4.2 (Orengo et al., 1997).

and DPLM-2 Wang et al. (2024b) also take a generative approach, with flow-based and diffusion language modeling. AIDO.Protein (Sun et al., 2024) is a 16 billion parameter pretrained protein language model that has been further adapted for inverse folding with conditional discrete diffusion language modeling objective. VFN-IF and VFN-IFE extend the Vector Field Network to inverse folding by using learnable vector computations over virtual atom coordinates (Mao et al., 2023).

## D  RETRIEVAL KERNEL

We model $\mathcal{R} = \{\mathcal{R}_i\}_{i=1}^{L}$ as a latent retrieval hypothesis. The kernel $p(\mathcal{R} \mid \mathcal{E}, D)$ admits both stochastic definitions and deterministic approximations.

**Stochastic retrieval.**  For residue $i$, let the cosine similarity between query embedding $\mathcal{E}_i$ and entity $d \in D$ (with embedding $\mathcal{E}(d)$) be

$$a_i(d) = \frac{\langle \mathcal{E}_i,\ \mathcal{E}(d) \rangle}{\|\mathcal{E}_i\|\,\|\mathcal{E}(d)\|}. \tag{14}$$

Convert to nonnegative weights using temperature $\tau > 0$ and normalize:

$$w_i(d) = \exp\big(a_i(d)/\tau\big), \qquad p_i(d) = \frac{w_i(d)}{\sum_{d' \in D} w_i(d')}. \tag{15}$$

Sample $K$ distinct entities $\mathcal{R}_i \subset D$ without replacement under a Plackett–Luce kernel. For an ordered $K$-tuple $\pi_i = (d_{i1}, \ldots, d_{iK})$ with distinct elements,

$$\Pr(\pi_i \mid \mathcal{E}_i, D) = \prod_{k=1}^{K} \frac{w_i(d_{ik})}{\displaystyle\sum_{d \in D \setminus \{d_{i1}, \ldots, d_{i,k-1}\}} w_i(d)}. \tag{16}$$

For the unordered set $\mathcal{R}_i$,

$$p(\mathcal{R}_i \mid \mathcal{E}_i, D) = \sum_{\pi_i \in \mathrm{Perm}(\mathcal{R}_i)} \prod_{k=1}^{K} \frac{w_i(d_{ik})}{\displaystyle\sum_{d \in D \setminus \{d_{i1}, \ldots, d_{i,k-1}\}} w_i(d)}. \tag{17}$$

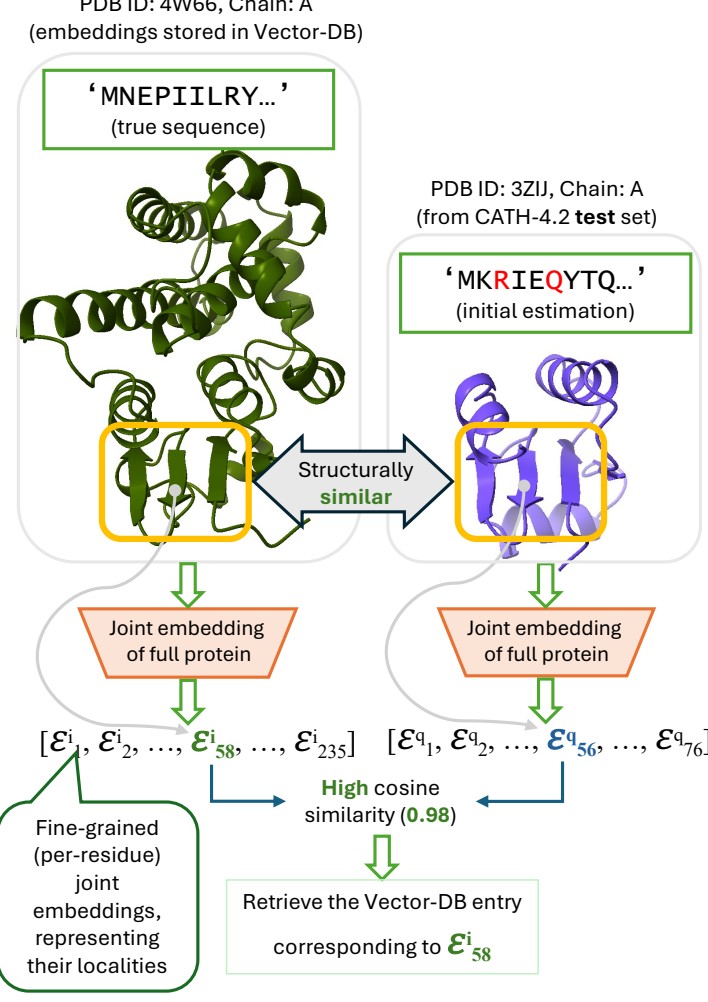

Figure 6: An example of how our vector DB search works. Here we leverage embedding $\mathcal{E}^i_j$ (representing the potential motif in residue $j$'s locality in protein $i$) to search representations of similar 3D localities in other proteins in the vector database to enrich context for later generation step.

The kernel factorizes across residues:

$$p(\mathcal{R} \mid \mathcal{E}, D) = \prod_{i=1}^{L} p(\mathcal{R}_i \mid \mathcal{E}_i, D), \qquad \mathcal{R} = \{\mathcal{R}_i\}_{i=1}^{L}. \tag{18}$$

We leverage this stochastic process (together with the full probabilistic model) when sampling diverse sequences (Appendix M).

**Deterministic approximation.** As $\tau \to 0$, Eq. 15 concentrates on maximizers of $a_i(d)$ and Eq. 16 sequentially selects the $K$ largest scores (ties broken arbitrarily). Thus $p(\mathcal{R}_i \mid \mathcal{E}_i, D)$ in Eq. 17 collapses to a point mass on the top-$K$ set:

$$\text{Top}K(\mathcal{E}_i; D) = \arg \max_{\substack{\mathcal{J} \subseteq D \\ |\mathcal{J}|=K}} \sum_{d \in \mathcal{J}} a_i(d). \tag{19}$$

Formally,

$$p(\mathcal{R} \mid \mathcal{E}, D) = \prod_{i=1}^{L} p(\mathcal{R}_i \mid \mathcal{E}_i, D), \quad p(\mathcal{R}_i \mid \mathcal{E}_i, D) = \delta\big(\mathcal{R}_i - \text{Top}K(\mathcal{E}_i; D)\big), \tag{20}$$

with Dirac distribution $\delta(\cdot)$; see Appendix D.1 for proof.

### D.1 CONVERGENCE OF STOCHASTIC RETRIEVAL TO DETERMINISTIC TOP-$K$

**Proposition 1.** *Let $a_i(d)$ denote the cosine similarity score between query embedding $\mathcal{E}_i$ and entity $d \in D$. Define weights and softmax probabilities*

$$w_i(d) = \exp\big(a_i(d)/\tau\big), \qquad p_i(d) = \frac{w_i(d)}{\sum_{d' \in D} w_i(d')}.$$

*Consider the Plackett–Luce sampler that draws an ordered $K$-tuple $\pi_i = (d_{i1}, \ldots, d_{iK})$ without replacement:*

$$\Pr(\pi_i \mid \mathcal{E}_i, D) = \prod_{k=1}^{K} \frac{w_i(d_{ik})}{\sum_{d \in D \setminus \{d_{i1}, \ldots, d_{i,k-1}\}} w_i(d)}.$$

*As $\tau \to 0$, the distribution over unordered retrieval sets $\mathcal{R}_i = \{d_{i1}, \ldots, d_{iK}\}$ converges to a point mass on the deterministic Top-$K$ set of scores $\mathrm{TopK}(\mathcal{E}_i; D)$, up to uniform randomness among exact ties.*

*Proof.* **Step 1: Single-choice limit.** Fix $i$ and write $a(d) := a_i(d)$. Let $M = \max_{d'} a(d')$ and $T = \{d : a(d) = M\}$ be the argmax set. Then

$$p_d(\tau) = \frac{e^{a(d)/\tau}}{\sum_{d'} e^{a(d')/\tau}} = \frac{e^{(a(d)-M)/\tau}}{\sum_{d'} e^{(a(d')-M)/\tau}}.$$

As $\tau \to 0$, the numerator converges to 1 if $d \in T$ and 0 otherwise. Hence

$$\lim_{\tau \to 0} p_d(\tau) = \begin{cases} 1/|T|, & d \in T, \\ 0, & d \notin T. \end{cases}$$

If $|T| = 1$, the maximizer $d^\star$ is selected with probability 1.

**Step 2: Sequential without replacement.** Plackett–Luce draws $K$ items by repeating the softmax on the remaining set. If $|T| = 1$, then $d_{i1} = d^\star$ w.p. 1. Removing $d^\star$, the argument applies inductively to the reduced set, so at each step the current maximum is selected. Thus the ordered tuple $\pi_i$ is the Top-$K$ scores in descending order.

If there are ties, the probability mass is split uniformly among tied maxima; once one is chosen, the argument recurses on the remaining set.

**Conclusion.** Therefore, for the unordered set $\mathcal{R}_i$,

$$\lim_{\tau \to 0} p(\mathcal{R}_i \mid \mathcal{E}_i, D) = \begin{cases} 1, & \mathcal{R}_i = \mathrm{TopK}(\mathcal{E}_i; D), \\ 0, & \text{otherwise}, \end{cases}$$

up to uniform randomness under exact ties. This proves the claim. $\square$

## E FROM LATENT MODEL TO TRAINING OBJECTIVE: ELBO, PRIOR–JENSEN BOUND, AND DETERMINISTIC REDUCTION

**Model recap.** Given backbone $B$ and database $D$, the latent variables are the embedding $\mathcal{E}$, the retrieval $\mathcal{R} = \{\mathcal{R}_i\}_{i=1}^{L}$, and the attribution $\mathbf{Z}$. The emission factorizes across residues with logits $\mathbf{Y}(\mathcal{E}, B, \mathcal{R})$:

$$p(\mathbf{S} \mid \mathbf{Z}, \mathcal{R}, \mathcal{E}, B) = \prod_{i=1}^{L} \mathrm{Cat}\Big(S_i; \mathrm{softmax}\big(\mathbf{Y}(\mathcal{E}, B, \mathcal{R})_i\big)\Big).$$

The joint and marginal are

$$p(\mathbf{S}, \mathbf{Z}, \mathcal{E}, \mathcal{R} \mid B, D) = p(\mathcal{E} \mid B)\, p(\mathcal{R} \mid \mathcal{E}, D)\, p(\mathbf{Z} \mid \mathcal{R}, \mathcal{E}, B)\, p(\mathbf{S} \mid \mathbf{Z}, \mathcal{R}, \mathcal{E}, B), \qquad (21)$$

$$p(\mathbf{S} \mid B, D) = \sum_{\mathbf{Z}, \mathcal{E}, \mathcal{R}} p(\mathcal{E} \mid B)\, p(\mathcal{R} \mid \mathcal{E}, D)\, p(\mathbf{Z} \mid \mathcal{R}, \mathcal{E}, B)\, p(\mathbf{S}, \mathbf{Z}, \mathcal{E}, \mathcal{R} \mid B, D). \qquad (22)$$

$$(23)$$

For notational simplicity, we use the summation symbol $\sum$ to denote marginalization over all latent variables, encompassing both summation (for discrete variables) and integration (for continuous embeddings).

### E.1 VARIATIONAL ELBO

**Theorem E.1** (Variational ELBO). *For any density $q(\mathcal{E}, \mathcal{R}, \mathbf{Z} \mid \mathbf{S}, B, D)$ with support contained in that of $p(\mathcal{E}, \mathcal{R}, \mathbf{Z} \mid B, D)$,*

$$\log p(\mathbf{S} \mid B, D) \geq \mathcal{L}_{\text{ELBO}}(q), \tag{24}$$

*where the ELBO can be written in either of the equivalent forms*

$$\mathcal{L}_{\text{ELBO}}(q) = \mathbb{E}_q\big[\log p(\mathbf{S} \mid \mathbf{Z}, \mathcal{R}, \mathcal{E}, B)\big] - \text{KL}\big(q(\mathcal{E}, \mathcal{R}, \mathbf{Z} \mid \mathbf{S}, B, D) \,\big\|\, p(\mathcal{E}, \mathcal{R}, \mathbf{Z} \mid B, D)\big), \tag{25}$$

$$= \mathbb{E}_q\big[\log p(\mathbf{S} \mid \mathbf{Z}, \mathcal{R}, \mathcal{E}, B)\big] + \mathbb{E}_q[\log p(\mathcal{E} \mid B)] + \mathbb{E}_q[\log p(\mathcal{R} \mid \mathcal{E}, D)]$$
$$+ \mathbb{E}_q[\log p(\mathbf{Z} \mid \mathcal{R}, \mathcal{E}, B)] - \mathbb{E}_q[\log q]. \tag{26}$$

*Moreover,*

$$\log p(\mathbf{S} \mid B, D) = \mathcal{L}_{\text{ELBO}}(q) + \text{KL}\big(q(\cdot) \,\|\, p(\mathcal{E}, \mathcal{R}, \mathbf{Z} \mid \mathbf{S}, B, D)\big), \tag{27}$$

*so that $-\log p(\mathbf{S} \mid B, D) \leq -\mathcal{L}_{\text{ELBO}}(q)$ (the negative ELBO upper-bounds the true NLL).*

*Proof.* Start from 23 and multiply and divide by $q(\mathcal{E}, \mathcal{R}, \mathbf{Z} \mid \mathbf{S}, B, D)$:

$$\log p(\mathbf{S} \mid B, D) = \log \sum_{\mathbf{Z}, \mathcal{E}, \mathcal{R}} q(\cdot) \frac{p(\mathbf{S}, \mathcal{E}, \mathcal{R}, \mathbf{Z} \mid B, D)}{q(\mathcal{E}, \mathcal{R}, \mathbf{Z} \mid \mathbf{S}, B, D)} = \log \mathbb{E}_q \left[ \frac{p(\mathbf{S}, \mathcal{E}, \mathcal{R}, \mathbf{Z} \mid B, D)}{q(\mathcal{E}, \mathcal{R}, \mathbf{Z} \mid \mathbf{S}, B, D)} \right].$$

By Jensen's inequality (concavity of log):

$$\log p(\mathbf{S} \mid B, D) = \log \mathbb{E}_q \left[ \frac{p(\mathbf{S}, \mathcal{E}, \mathcal{R}, \mathbf{Z} \mid B, D)}{q(\mathcal{E}, \mathcal{R}, \mathbf{Z} \mid \mathbf{S}, B, D)} \right] \geq \mathbb{E}_q \left[ \log \frac{p(\mathbf{S}, \mathcal{E}, \mathcal{R}, \mathbf{Z} \mid B, D)}{q(\mathcal{E}, \mathcal{R}, \mathbf{Z} \mid \mathbf{S}, B, D)} \right]$$
$$= \mathbb{E}_q[\log p(\mathbf{S}, \mathcal{E}, \mathcal{R}, \mathbf{Z} \mid B, D)] - \mathbb{E}_q[\log q].$$

Expanding the joint via the model factorization gives 26, and grouping terms yields 25. For the decomposition with the posterior, observe by Bayes:

$$p(\mathcal{E}, \mathcal{R}, \mathbf{Z} \mid \mathbf{S}, B, D) = \frac{p(\mathbf{S}, \mathcal{E}, \mathcal{R}, \mathbf{Z} \mid B, D)}{p(\mathbf{S} \mid B, D)}.$$

Hence

$$\text{KL}\big(q \,\|\, p(\cdot \mid \mathbf{S}, B, D)\big) = \mathbb{E}_q \left[ \log \frac{q}{p(\cdot \mid \mathbf{S}, B, D)} \right]$$
$$= \mathbb{E}_q[\log q] - \mathbb{E}_q[\log p(\mathbf{S}, \mathcal{E}, \mathcal{R}, \mathbf{Z} \mid B, D)] + \log p(\mathbf{S} \mid B, D),$$

i.e.

$$\log p(\mathbf{S} \mid B, D) = \underbrace{\mathbb{E}_q[\log p(\mathbf{S}, \mathcal{E}, \mathcal{R}, \mathbf{Z} \mid B, D)] - \mathbb{E}_q[\log q]}_{\mathcal{L}_{\text{ELBO}}(q)} + \text{KL}\big(q \,\|\, p(\cdot \mid \mathbf{S}, B, D)\big).$$

Since $\text{KL} \geq 0$, the inequality follows. $\square$

### E.2 TRAINABLE RETRIEVAL KERNEL

Under the factorizations of the amortized posterior

$$q(\mathcal{E}, \mathcal{R}, \mathbf{Z} \mid \mathbf{S}, B, D) = q(\mathcal{E} \mid \mathbf{S}, B) \, q(\mathcal{R} \mid \mathbf{S}, \mathcal{E}, D) \, q(\mathbf{Z} \mid \mathbf{S}, \mathcal{R}, \mathcal{E}, B),$$

and the prior

$$p(\mathcal{E}, \mathcal{R}, \mathbf{Z} \mid B, D) = p(\mathcal{E} \mid B) \, p(\mathcal{R} \mid \mathcal{E}, D) \, p(\mathbf{Z} \mid \mathcal{R}, \mathcal{E}, B),$$

the exact ELBO from Eq. 25 becomes

$$\mathcal{L}_{\text{ELBO}}(q) = \mathbb{E}_q[\log p(\mathbf{S} \mid \mathbf{Z}, \mathcal{R}, \mathcal{E}, B)]$$
$$- \text{KL}(q(\mathcal{E} \mid \mathbf{S}, B) \,\|\, p(\mathcal{E} \mid B))$$
$$- \mathbb{E}_q[\text{KL}(q(\mathcal{R} \mid \mathbf{S}, \mathcal{E}, D) \,\|\, p(\mathcal{R} \mid \mathcal{E}, D))]$$
$$- \mathbb{E}_q[\text{KL}(q(\mathbf{Z} \mid \mathbf{S}, \mathcal{R}, \mathcal{E}, B) \,\|\, p(\mathbf{Z} \mid \mathcal{R}, \mathcal{E}, B))].$$

With $q(\mathcal{E} \mid \mathbf{S}, B) = p(\mathcal{E} \mid B)$ and $q(\mathbf{Z} \mid \mathbf{S}, \mathcal{R}, \mathcal{E}, B) = p(\mathbf{Z} \mid \mathcal{R}, \mathcal{E}, B)$,

$$\mathcal{L}_{\text{ELBO}}(q) = \mathbb{E}_q\big[\log p(\mathbf{S} \mid \mathbf{Z}, \mathcal{R}, \mathcal{E}, B)\big] - \mathbb{E}_q\big[\text{KL}\big(q(\mathcal{R} \mid \mathbf{S}, \mathcal{E}, D) \,\|\, p(\mathcal{R} \mid \mathcal{E}, D)\big)\big],$$

which enables training the retrieval prior $p(\mathcal{R} \mid \mathcal{E}, D)$ through minimization of the KL-divergence.

### E.3 PRIOR–JENSEN LOWER BOUND

**Corollary E.1.1** (Prior–Jensen bound). *Let $p(\mathcal{E}, \mathcal{R}, \mathbf{Z} \mid B, D)$ be the latent prior induced by the model. Then*

$$\log p(\mathbf{S} \mid B, D) \geq \mathbb{E}_{p(\mathcal{E}, \mathcal{R}, \mathbf{Z} \mid B, D)}\big[\log p(\mathbf{S} \mid \mathbf{Z}, \mathcal{R}, \mathcal{E}, B)\big], \tag{28}$$

*equivalently*

$$-\log p(\mathbf{S} \mid B, D) \leq -\mathbb{E}_p\big[\log p(\mathbf{S} \mid \mathbf{Z}, \mathcal{R}, \mathcal{E}, B)\big].$$

*Proof.* Take $q(\mathcal{E}, \mathcal{R}, \mathbf{Z} \mid \mathbf{S}, B, D) = p(\mathcal{E}, \mathcal{R}, \mathbf{Z} \mid B, D)$ in Theorem E.1. Then $\mathrm{KL}(q\|p(\cdot \mid B, D)) = 0$ in 25, and the ELBO reduces to $\mathbb{E}_p[\log p(\mathbf{S} \mid \mathbf{Z}, \mathcal{R}, \mathcal{E}, B)]$, which is therefore a lower bound on $\log p(\mathbf{S} \mid B, D)$. Alternatively, apply Jensen directly to

$$\log p(\mathbf{S} \mid B, D) = \log \mathbb{E}_{p(\mathcal{E}, \mathcal{R}, \mathbf{Z} \mid B, D)}\big[p(\mathbf{S} \mid \mathbf{Z}, \mathcal{R}, \mathcal{E}, B)\big] \geq \mathbb{E}_p\big[\log p(\mathbf{S} \mid \cdot)\big].$$

$\square$

### E.4 DETERMINISTIC REDUCTION AND TIGHTNESS

**Proposition 2** (Deterministic reduction and tightness). *Assume*

$$p(\mathcal{E} \mid B) = \delta\big(\mathcal{E} - \hat{\mathcal{E}}^q\big), \quad p(\mathcal{R} \mid \mathcal{E}, D) = \prod_{i=1}^{L} \delta\big(\mathcal{R}_i - \mathrm{Top}K(\hat{\mathcal{E}}_i^q; D)\big),$$

$$p(\mathbf{Z} \mid \mathcal{R}, \mathcal{E}, B) = \delta\big(\mathbf{Z} - \mathcal{A}(\hat{\mathcal{E}}^q, B^q, \mathcal{R})\big).$$

*Let $\mathcal{R}^\star = \{\mathrm{Top}K(\hat{\mathcal{E}}_i^q; D)\}_i$ and $\mathbf{Z}^\star = \mathcal{A}(\hat{\mathcal{E}}^q, B, \mathcal{R}^\star)$. Then*

$$\log p\big(\mathbf{S} \mid B^q, D\big) = \log p\big(\mathbf{S} \mid \mathbf{Z}^\star, \mathcal{R}^\star, \hat{\mathcal{E}}^q, B^q\big), \tag{29}$$

*and the right hand side equals the standard per-residue log-likelihood*

$$\log p\big(\mathbf{S} \mid \mathbf{Z}^\star, \mathcal{R}^\star, \hat{\mathcal{E}}^q, B^q\big) = \sum_{i=1}^{L} \log \mathrm{softmax}\big(\mathbf{Y}(\hat{\mathcal{E}}^q, B^q, \mathcal{R}^\star)_i\big)_{S_i}. \tag{30}$$

*Consequently,*

$$-\log p(\mathbf{S} \mid B, D) = -\sum_{i=1}^{L} \log \mathrm{softmax}\big(\mathbf{Y}(\hat{\mathcal{E}}^q, B^q, \mathcal{R}^\star)_i\big)_{S_i}.$$

*Proof.* Under the stated Dirac measures, the summations in 23 collapse to the single configuration $(\hat{\mathcal{E}}^q, \mathcal{R}^\star, \mathbf{Z}^\star)$:

$$p(\mathbf{S} \mid B, D) = p\big(\mathbf{S} \mid \mathbf{Z}^\star, \mathcal{R}^\star, \hat{\mathcal{E}}^q, B^q\big).$$

By the emission factorization, this conditional equals $\prod_{i=1}^{L} \mathrm{Cat}(S_i; \mathrm{softmax}(\mathbf{Y}(\hat{\mathcal{E}}^q, B^q, \mathcal{R}^\star)_i))$. Taking logs yields the stated sum of per-residue log-softmax terms. $\square$

### E.5 CONSEQUENCES FOR THE TRAINING OBJECTIVE

Combining Corollary E.1.1 and Proposition 2:

$$\log p(\mathbf{S} \mid B, D) \geq \mathbb{E}_p\big[\log p(\mathbf{S} \mid \mathbf{Z}, \mathcal{R}, \mathcal{E}, B)\big] \text{ and } \log p(\mathbf{S} \mid B, D) = \log p\big(\mathbf{S} \mid \mathbf{Z}^\star, \mathcal{R}^\star, \hat{\mathcal{E}}^q, B^q\big),$$

so the *plug-in* negative log-likelihood

$$\mathcal{L}_{\mathrm{NLL}} = -\sum_{i=1}^{L} \log \mathrm{softmax}\big(\mathbf{Y}(\hat{\mathcal{E}}^q, B^q, \mathcal{R}^\star)_i\big)_{S_i}$$

is (i) exactly the NLL of the deterministic latent model, and (ii) an *upper bound* on the true NLL of the stochastic latent model:

$$-\log p(\mathbf{S} \mid B, D) \leq \mathcal{L}_{\mathrm{NLL}}.$$

Equality holds when the stochastic latents degenerate to Dirac measures (our current design), or when a variational posterior collapses to a point mass concentrated at $(\hat{\mathcal{E}}^q, \mathcal{R}^\star, \mathbf{Z}^\star)$.

### E.6 DERIVED PROBABILISTIC MODEL

Substituting the kernels into Eq. 3 gives

$$p(\mathbf{S} \mid B, D) = \sum_{\mathcal{E},\mathcal{R},\mathbf{Z}} \Big[ \prod_{i=1}^{L} p(\mathcal{E}_i \mid B) \sum_{\pi_i \in \mathrm{Perm}(\mathcal{R}_i)} \prod_{k=1}^{K} \frac{w_i(d_{ik})}{\sum_{d \in D \setminus \{d_{i1}, \ldots, d_{i,k-1}\}} w_i(d)} \Big]$$

$$p(\mathbf{Z} \mid \mathcal{R}, \mathcal{E}, B) \prod_{i=1}^{L} \mathrm{Cat}\big(S_i; \mathrm{softmax}(\mathbf{Y}(\mathcal{E}, B, \mathcal{R}))_i\big). \quad (31)$$

Under the deterministic approximations,

$$p(\mathbf{S} \mid B, D) = \sum_{\mathcal{R},\mathbf{Z}} \Big[ \prod_{i=1}^{L} \delta\big(\mathcal{R}_i - \mathrm{Top}K(\hat{\mathcal{E}}_i^q; D)\big) \Big] \delta\big(\mathbf{Z} - \mathcal{A}(\hat{\mathcal{E}}^q, B, \mathcal{R})\big)$$

$$\prod_{i=1}^{L} \mathrm{Cat}\big(S_i; \mathrm{softmax}(\mathbf{Y}(\hat{\mathcal{E}}^q, B, \mathcal{R}))_i\big). \quad (32)$$

### E.7 TRAINING OBJECTIVE UNDER DETERMINISTIC REDUCTION

We target the true marginal $\log p(\mathbf{S} \mid B, D)$ and optimize its *prior–Jensen lower bound* (formal proof in App. E):

$$\log p(\mathbf{S} \mid B, D) \;\geq\; \mathbb{E}_p\Big[ \log p(\mathbf{S} \mid \cdot) \Big] = \mathbb{E}_p\Big[ \sum_{i=1}^{L} \log \mathrm{Cat}\big(S_i; \mathrm{softmax}\, \mathbf{Y}_i(\mathcal{E}, B, \mathcal{R}; \theta)\big) \Big]. \quad (33)$$

Equivalently, we minimize corresponding Jensen *negative ELBO* (NELBO) to learn parameter set $\theta$:

$$\hat{\theta} = \arg\min_{\theta} \; \mathbb{E}_{p(\mathcal{E},\mathcal{R},\mathbf{Z}|B,D)}\Big[ \sum_{i=1}^{L} \log \mathrm{Cat}\big(S_i; \mathrm{softmax}\, \mathbf{Y}_i(\mathcal{E}, B, \mathcal{R}; \theta)\big) \Big], \quad (34)$$

Under our *deterministic reduction* for any query protein $B^q$ (App. E, Prop. 2) with $\mathcal{E} = \hat{\mathcal{E}}^q$, $\mathcal{R}^\star = \mathrm{Top}K(\hat{\mathcal{E}}^q; D)$, $\mathbf{Z}^\star = \mathcal{A}(\hat{\mathcal{E}}^q, B^q, \mathcal{R}^\star)$, the *prior-Jensen* lower bound in Eq. 28 is *tight* and the objective collapses to standard per-residue cross-entropy:

$$\hat{\theta} = \arg\min_{\theta} \left[ -\sum_{i=1}^{L} \log \mathrm{softmax}\big(\mathbf{Y}(\hat{\mathcal{E}}^q, B^q, \mathcal{R}^\star; \theta)_i\big)_{S_i} \right], \quad (35)$$

with gradients flowing through the learnable parameters $\theta = \{\theta_{\mathbf{Z}}, \theta_B\}$; the retrieval $\mathrm{Top}K$ is treated as fixed and non-differentiable in this deterministic setting.

## F AGGREGATION AND GENERATION WITH OUR DECODER

We aggregate the retrieved entities $\mathcal{R}^\star$ to generate a new sequence $\tilde{S}^q$. We do this with a series of $T$ consecutive learnable blocks, each consisting of one multihead self-attention layer (MHSA), one multihead cross-attention layer (MHCA), and two bottleneck multilayer perceptrons (Houlsby et al., 2019) ($T$ is a hyperparameter). In the rest of this article, we refer to this hybrid block as multihead self-cross attention block (MHSCA).

As shown in Fig. 2 (Point ⑤) and Fig. 7, for $\forall l \in [1, |\hat{S}^q|]$ we first extract the embedding vectors $\{(\mathcal{E}_j^i)_k : k \in [1, \tilde{K}]\}$ from our retrieved entities. We then merge them with the query vector $\mathcal{E}_l^q$ and linearly project the output to create a matrix $\mathcal{H}_l^q \in \mathbb{R}^{(\tilde{K}+1) \times d'}$ as,

$$\mathcal{H}_l^q = \mathrm{concat}(\{(\mathcal{E}_j^i)_k : k \in [1, \tilde{K}]\} \cup \{\mathcal{E}_l^q\}) \, W_{\mathcal{H}}, \quad (36)$$

where concat(.) performs a concatenation operation on the vectors in the union set, $W_{\mathcal{H}} \in \mathbb{R}^{d \times d'}$ is a learnable parameter, and $d'$ is the output embedding dimension. Then for the whole query

protein $P^q$ we get a tensor $\mathcal{H}^q = [\mathcal{H}_1^q, \mathcal{H}_2^q, \ldots, \mathcal{H}_{|\hat{S}^q|}^q] \in \mathbb{R}^{|\hat{S}^q| \times (\tilde{K}+1) \times d'}$, which is used as *query*, *key*, and *value* of MHSA (see Vaswani (2017) for definitions). To ensure that the generator can effectively leverage any residual 3D structural information, we also encode the input structure $B^q$ separately using a structural encoder, where *no sequence* information is provided. Similar to Sun et al. (2024), we leverage ProteinMPNN-CMLM (Zheng et al., 2023) for structure encoding, which is a variant of the original ProteinMPNN method (Dauparas et al., 2022) trained with conditional masked language modeling objective (Ghazvininejad et al., 2019). This generates structural encoding $\rho^q \in \mathbb{R}^{|\hat{S}^q| \times d^\rho}$. This encoding is then linearly transformed and merged with a linear projection of query encoding $\mathcal{E}^q$, creating a new representation matrix $\theta^q \in \mathbb{R}^{|\hat{S}^q| \times d'}$, where each element $\theta_l^q =$ concat($\{\rho_l^q W_\rho, \mathcal{E}_l^q W_\mathcal{E}\}$) $\in \mathbb{R}^{d'}$, with two learnable parameters $W_\rho \in \mathbb{R}^{d^\rho \times \frac{d'}{2}}$ and $W_\mathcal{E} \in \mathbb{R}^{d \times \frac{d'}{2}}$. For our MHCA blocks, we use $\theta^q$ as the *query*, and $\mathcal{H}^q$ as both the *key* and *value*. The motivation behind such design of MHSCA is, while MHSA layers can help jointly attend to multiple parts of the input protein as well as their corresponding retrieved embeddings, MHCA can help extract any kind of residual structural information needed to better decode the sequence. Moreover, since the MHCA here preserves the same dimension as $\theta^q$, the output representation has $|\hat{S}^q|$ vectors which we can directly pass through another linear layer to generate the output logits $\mathbf{Y} \in \mathbb{R}^{|\hat{S}^q| \times d'}$. Sampling with $\mathbf{Y}$ provides us with a newly generated sequence $\tilde{S}^q$.

## G POST-HOC MEMORY-GROWTH ANALYSIS

In our design, each residue embedding $\mathcal{E}_r^p$ summarizes the *local motif* (or potential motif) around residue $r$ in protein $p$. Let $\phi(\cdot)$ map a residue neighborhood to a motif representation in a metric space $(\mathcal{M}, d)$, and let the database (memory) be $D = \{d = (\mathcal{E}_r^p, r, p)\}$. At inference, for each query residue $i$ we retrieve neighbors in $D$ by similarity of $\mathcal{E}_i$ to $\mathcal{E}(d)$, which effectively searches for nearby motifs in $\mathcal{M}$.

---

**Definition G.1** (Motif $\varepsilon$-coverage). *For tolerance $\varepsilon > 0$ and query distribution $\mathcal{Q}$ over motifs, the* coverage *of $D$ is* $\mathrm{Cov}_\varepsilon(D) = \mathrm{Pr}_{m \sim \mathcal{Q}} \big[ \min_{d \in D} d\big(m, \phi(d)\big) \leq \varepsilon \big]$.

---

**Proposition 3** (Coverage saturation). *Assume i.i.d. sampling of database motifs from the same distribution $\mathcal{Q}$ as test queries, and that the motif space admits a finite $\varepsilon$-cover number $\mathcal{N}_\varepsilon < \infty$. Then $\mathrm{Cov}_\varepsilon(D_n) \to 1$ as $|D_n| \to \infty$, and the expected marginal coverage gain from adding a batch of $k$ new entries satisfies $\mathbb{E}\big[\mathrm{Cov}_\varepsilon(D_{n+k}) - \mathrm{Cov}_\varepsilon(D_n)\big] = \mathcal{O}\big((1 - \frac{1}{\mathcal{N}_\varepsilon})^n\big)$.*

*Intuition.* Each database item covers an $\varepsilon$-ball in $\mathcal{M}$. Under i.i.d. sampling, uncovered mass shrinks geometrically with $n$ until most query motifs lie within $\varepsilon$ of at least one memory item. Beyond this point, additional samples mostly fall into already covered regions, yielding negligible retrieval improvements and thus minimal downstream gains. $\square$

**Implication.** Because each residue embedding encodes its local motif, a sufficiently large memory $D$ achieves high $\varepsilon$-coverage of the motif space. Once coverage saturates, top-$K$ neighbors (or their stochastic variants) are already near-optimal, so *post-hoc* accretion of similar residues contributes little to retrieval quality or sequence recovery, absent targeted diversification or retraining.

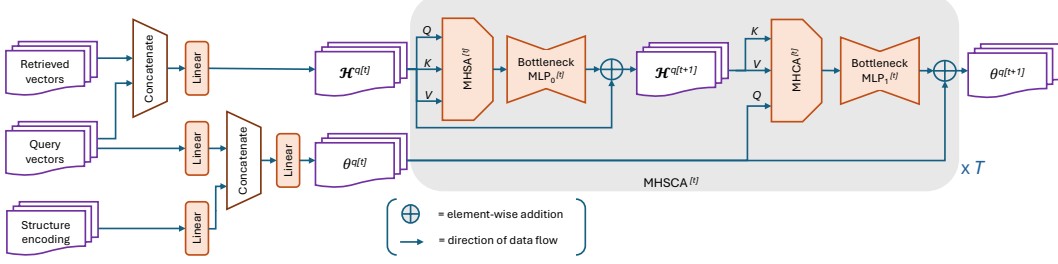

Figure 7: Our aggregation and generation module. It uses $T$ consecutive blocks of *Multi-Head Self-Cross Attention* (MHSCA). Here the super-script "$[t]$" corresponds to the *index* of the current MHSCA block (and also its components and their inputs). See Sec. F for details.

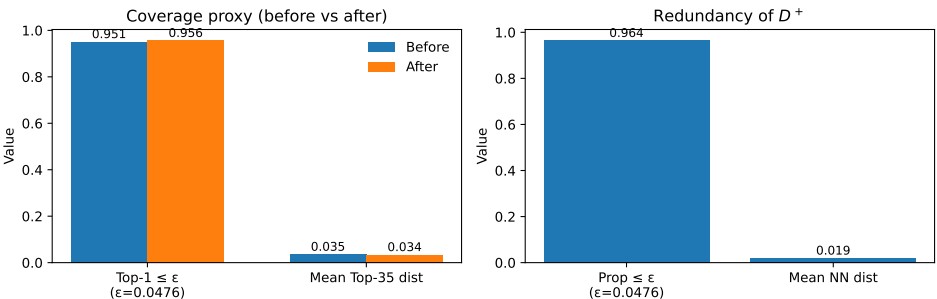

Figure 8: Post-hoc memory growth diagnostics. *Left*: coverage proxy before/after adding $D^+$—fraction of queries with Top-1 cosine distance $\leq \varepsilon$ ($\varepsilon = 0.0476$) and mean Top-35 distance. Coverage rises only slightly ($0.951 \rightarrow 0.956$) and mean Top-35 distance improves marginally ($0.0355 \rightarrow 0.0340$), indicating near-saturation. *Right*: redundancy of the added memory $D^+$ measured by nearest-neighbor distance to the original memory $D$. The vast majority (96.4%) of new entries already lie within $\varepsilon$ of an existing item (mean NN distance 0.0192), explaining the negligible coverage gain.

### G.1 DIAGNOSTICS

**Does enlarging the memory post-hoc help?**  To test the hypothesis that our memory already $\varepsilon$-covers most query motifs, we augment the fixed memory $D$ with a small disjoint batch $D^+$ from newer PDB entries [1] (no parameter updates) and re-index. We report two diagnostics:

**Coverage proxy.** For each query residue embedding $\mathcal{E}_i$, we compute the cosine distance of its nearest neighbor in the memory, $d_{\cos}^{(1)}(i; D)$, and the mean of its top-$K$ distances, $\bar{d}_{\cos}^{(K)}(i; D)$. We summarize by the fraction of queries with $d_{\cos}^{(1)}(i; D) \leq \varepsilon$ and by $\mathbb{E}_i[\bar{d}_{\cos}^{(K)}(i; D)]$, where $\varepsilon$ is fixed to the $q$-th percentile of $d_{\cos}^{(1)}(i; D)$ on the *original* memory (e.g., $q{=}95$).

**Redundancy of $D^+$.** For each new memory item $d \in D^+$, we compute its nearest-neighbor cosine distance to the original memory $D$, $d_{\cos}^{\mathrm{NN}}(d; D)$. We report the proportion with $d_{\cos}^{\mathrm{NN}}(d; D) \leq \varepsilon$ and the mean $\mathbb{E}_{d \in D^+}[d_{\cos}^{\mathrm{NN}}(d; D)]$.

**Finding.** We tested our hypothesis on the TS50 test set. The results are depicted in Fig. 8. The fraction of queries with a nearest neighbor within $\varepsilon = 0.0476$ cosine distance increased only marginally from $95.1\%$ to $95.6\%$, indicating that coverage was already near saturation. Similarly, the mean Top-35 cosine distance improved slightly ($0.0355 \rightarrow 0.0340$), a negligible gain given the scale of retrieval noise. By contrast, the added entries themselves were highly redundant: $96.4\%$ of $D^+$ items had a nearest neighbor within $\varepsilon$ in the original memory, with a mean nearest-neighbor distance of 0.0192. These results confirm that post-hoc memory growth mostly contributes redundant motifs and provides no meaningful benefit for retrieval or sequence recovery. This supports our design choice to treat the vector database as a fixed prior-knowledge memory.

**Detailed ablation.**  For completeness, we ablate the impact of database size by comparing three PRISM configurations: (i) $D$ constructed from the CATH-4.2 training set, (ii) $D^+$ built from new PDB entries, and (iii) their union. The results in Tab. 13 show that all variants achieve virtually indistinguishable perplexity and recovery scores, with differences well within statistical noise. For example, on CAMEO 2022 the AAR stabilizes at ~64.6% across all database choices, and on the CATH-4.2 test and validation splits the gap remains below $0.2\%$. These findings suggest that the CATH-4.2–based database already provides near-complete motif coverage, and that additional PDB entries primarily add redundant fragments rather than new information. This empirical evidence supports our design decision to fix the database as a compact, prior-knowledge memory: it ensures efficiency while preserving accuracy.

---

[1] all samples form the PBD split here: https://zenodo.org/records/15424801

Table 13: Ablation on the size of database. Results are rounded to two decimal points, hence very small changes are not reflected.

| Models | CAMEO 2022 | | CATH-4.2 test split | | CATH-4.2 val split | |
|---|---|---|---|---|---|---|
| | PPL ↓ | AAR % ↑ | PPL ↓ | AAR % ↑ | PPL ↓ | AAR % ↑ |
| Base estimator (AIDO.Protein-IF) | 2.68 | 63.52 (60.56/64.17) | 2.94 | 58.60 (57.27/60.13) | 2.90 | 58.73 (58.00/60.62) |
| PRISM (VDB: CATH 4.2 train) | 2.53 | 64.63 (61.30/64.81) | 2.71 | 60.43 (58.55/61.41) | 2.68 | 60.26 (59.28/61.89) |
| PRISM (VDB: PDB new) | 2.53 | 64.67 (61.25/64.81) | 2.71 | 60.23 (58.56/61.41) | 2.68 | 60.26 (59.29/61.91) |
| PRISM (VDB: CATH 4.2 train + PDB new) | 2.53 | 64.67 (61.27/64.82) | 2.71 | 60.43 (58.56/61.41) | 2.68 | 60.26 (59.29/61.90) |

## H  LENGTH VS. RECOVERY

To further examine how model performance scales with protein length, we stratified the CATH-4.2 test set into length bins and compared amino-acid recovery rates (AAR) between AIDO.Protein-IF and PRISM. As shown in Fig. 4, both models exhibit the expected trend of improved recovery with increasing sequence length, reflecting richer structural context in longer backbones. Crucially, across all bins, the distribution of PRISM's recovery rates consistently shifts upward relative to AIDO.Protein-IF, indicating that the gains are not restricted to a narrow subset of proteins but hold robustly across varying sequence lengths. Notably, in the shorter length regimes ($< 200$ residues), where inverse folding is traditionally more challenging, PRISM delivers marked improvements in both median and interquartile range, suggesting that fine-grained retrieval particularly benefits proteins with limited contextual information. At larger lengths ($> 300$ residues), the advantage remains evident, with PRISM maintaining higher medians and tighter variability, underscoring its scalability. This distributional analysis complements the average recovery metrics and highlights that PRISM achieves *consistent, robust gains across protein lengths*, reinforcing its generality beyond aggregate statistics.

## I  FOLDABILITY ANALYSIS

In Appendix Tab. 14, we provide additional foldability analysis results on TS50 an TS500 datasets.

Table 14: Foldability comparison using AF2 protein folding model. The median and the mean are provided outside and inside the parenthesis, respectively.

| Models | TS50 | | | TS500 | | |
|---|---|---|---|---|---|---|
| | RMSD ↓ | sc-TM ↑ | pLDDT ↑ | RMSD ↓ | sc-TM ↑ | pLDDT ↑ |
| AIDO.Protein-IF | 1.075 (1.2) | 0.956 (0.938) | 0.949 (0.937) | 1.18 (1.372) | 0.96 (0.904) | 0.951 (**0.931**) |
| PRISM (ours) | **0.985 (1.13)** | **0.964 (0.943)** | **0.95 (0.939)** | **1.125 (1.351)** | **0.964 (0.905)** | **0.952** (0.929) |

## J  CONTRIBUTION OF HYBRID DECODER WITH MHSCA

In Appendix Tab. 15 we provide further ablation on hybrid-attention vs cross-attn-only on test sets CAMEO 2022 and PDB date split.

Table 15: Ablation on hybrid-attention vs cross-attn-only (CAMEO 2022 and PDB date split).

| Models | CAMEO 2022 | | PDB date split | |
|---|---|---|---|---|
| | PPL ↓ | AAR % ↑ | PPL ↓ | AAR % ↑ |
| PRISM (w/o MHSA) | 2.60 | **64.63** (60.39/64.07) | 2.43 | 66.67 (66.56/67.77) |
| PRISM (full) | **2.53** | **64.63** (61.30/64.81) | **2.35** | **67.47** (67.37/68.51) |

## K  CONTRIBUTION OF AGGREGATION DEPTH

In Appendix Tab. 16 we provide additional ablation results on the contribution of aggregation depth, i.e., the impact of the number of MHSCA blocks on the test sets CAMEO 2022 and PDB date split.

Table 16: Ablation on the number of MHSCA blocks (CAMEO 2022 and PDB date split).

| # of blocks | CAMEO 2022 | | PDB date split | |
|---|---|---|---|---|
| | PPL ↓ | AAR % ↑ | PPL ↓ | AAR % ↑ |
| N/A (base est.) | 2.68 | 63.52 (60.56/64.17) | 2.49 | 66.27 (66.37/67.64) |
| 1 | 2.54 | **64.67** (61.31/64.81) | 2.36 | 67.20 (67.33/68.49) |
| 2 | **2.53** | 64.63 (61.30/64.81) | **2.35** | **67.47** (67.37/68.51) |
| 3 | 2.54 | 64.61 (61.27/64.80) | 2.36 | 67.41 (67.33/68.47) |

## L  SEQUENCE RECOVERY AND DESIGNABILITY ANALYSIS

 Appendix Tab. 17 reports additional results on sequence recovery and designability for the CATH-4.2 test split. We follow the standard evaluation protocol used in Mao et al. (2023) and Yim et al. (2023), computing the proportions of generated sequences whose ESMFold (Lin et al., 2023) structures satisfy the designability criteria: scTM > 0.5 (%) and scRMSD < 2 (%). Here we report scores for two PRISM variants: (i) using the original frozen joint encoder $\mathcal{G}$ (denoted "orig. $\mathcal{G}$"), and (ii) using fine-tuned $\mathcal{G}$ and iterative refinement (denoted "ft. $\mathcal{G}$, iter."). The results show that PRISM substantially improves both sequence recovery and structural designability, achieving state-of-the-art performance across all metrics.

Table 17: Performance comparison on the CATH-4.2 test split for sequence recovery and designability. All baseline scores are reported from Mao et al. (2023). Best and second-best scores are shown in bold and italic, respectively. Here subscript $aido$ denotes that AIDO.Protein-IF was used as the base estimator.

| Model | PPL ↓ | AAR (%) ↑ | scTM > 0.5 (%) ↑ | scRMSD < 2 (%) ↑ |
|---|---|---|---|---|
| PiFold | 4.55 | 51.66 | 90.98 | 60.35 |
| LM-Design | 4.41 | 54.41 | 89.42 | 58.41 |
| VFN-IF | 4.17 | 54.74 | 92.37 | 62.89 |
| VFN-IFE | 3.36 | *62.67* | 93.29 | 64.16 |
| PRISM$_{aido}$ (orig. $\mathcal{G}$) | *2.71* | 60.43 | *94.82* | **69.73** |
| PRISM$_{aido}$ (ft. $\mathcal{G}$, iter.) | **2.65** | **63.33** | **95.36** | *69.55* |

## M  RECOVERY–DIVERSITY TRADE-OFF VIA TEMPERATURE SAMPLING

An important question for inverse folding is whether improvements in recovery come at the cost of reduced sequence diversity, since a practical design framework must balance both fidelity to the native sequence and exploration of alternative solutions. To probe this trade-off, we performed controlled sampling experiments with PRISM by varying the decoding temperature while holding all other factors fixed. For each backbone, we generated 100 candidate sequences at temperatures ranging from 0.1 to 1.3 and evaluated (i) *Recovery Rate*, measured as mean sequence identity to the native, and (ii) *Diversity*, measured as $1-$ average pairwise identity (PID) among the sampled sequences.

**Recovery–Diversity Frontier.**  Fig. 9 (left) illustrates the recovery–diversity frontier achieved by PRISM. At very low temperatures (e.g., $T = 0.1$), the model collapses to near-deterministic decoding, yielding high recovery ($\sim$0.58) but very limited diversity ($\sim$0.06). As the temperature increases, diversity rises monotonically, reaching 0.63 at $T = 1.3$. Importantly, this comes with only a gradual reduction in recovery, which remains above 0.40 even at the highest temperatures. This smooth frontier indicates that PRISM does not degenerate into trivial random sampling; instead, it maintains a meaningful distributional match to the native even under exploratory sampling.

**Recovery vs. Temperature.**  The middle panel confirms that recovery declines as temperature increases, consistent with expectations that flatter distributions produce more varied but less native-like sequences (Fig. 9 (middle)). However, the slope of this decline is shallow: from $T = 0.1$ to $T = 1.3$, recovery drops by only $\sim$0.16 absolute. This robustness suggests that the retrieval-augmented architecture sharpens conditional probabilities enough to preserve signal even under stochastic decoding.

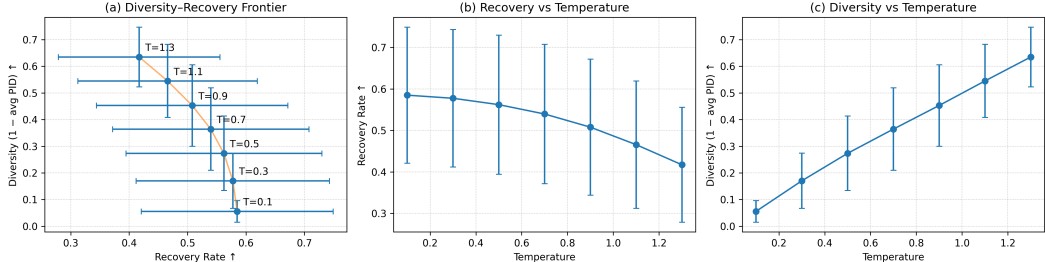

Figure 9: Recovery–diversity trade-off via temperature sampling. (a) *Diversity–Recovery frontier*: PRISM maintains high recovery while offering controllable diversity as temperature increases. (b) *Recovery vs. temperature*: recovery decreases gradually under stochastic sampling, demonstrating robustness. (c) *Diversity vs. temperature*: diversity increases nearly linearly, enabling rich alternative designs. Together, these results highlight PRISM's ability to support a tunable accuracy–diversity trade-off without collapse.

Table 18: Comparison of Base Estimator (AIDO.Protein-IF) and PRISM across multiple benchmarks. We report perplexity (PPL, lower is better) and amino acid recovery (AAR, higher is better), along with absolute and percentage improvements of PRISM over the base model.

| Models | TS50 | | TS500 | | CAMEO 2022 | | PDB date split | | CATH-4.2 test split | | CATH-4.2 val split | | Average | |
|---|---|---|---|---|---|---|---|---|---|---|---|---|---|---|
| | PPL ↓ | AAR % ↑ | PPL ↓ | AAR % ↑ | PPL ↓ | AAR % ↑ | PPL ↓ | AAR % ↑ | PPL ↓ | AAR % ↑ | PPL ↓ | AAR % ↑ | PPL ↓ | AAR % ↑ |
| Base estimator (AIDO.Protein-IF) | 2.68 | 66.19 | 2.42 | 69.66 | 2.68 | 63.52 | 2.49 | 66.27 | 2.94 | 58.60 | 2.90 | 58.73 | 2.685 | 63.83 |
| **PRISM (full)** | 2.43 | 67.92 | 2.27 | 70.53 | 2.53 | 64.63 | 2.35 | 67.47 | 2.71 | 60.43 | 2.68 | 60.26 | 2.495 | 65.87 |
| Absolute change Δ | -0.25 | +1.73 | -0.15 | +0.87 | -0.15 | +1.11 | -0.14 | +1.20 | -0.23 | +1.83 | -0.22 | +1.53 | -0.19 | +2.04 |
| Relative change Δ% | -9.3% | +2.6% | -6.2% | +1.2% | -5.6% | +1.8% | -5.6% | +1.8% | -7.8% | +3.1% | -7.6% | +2.6% | -7.1% | +3.2% |

**Diversity vs. Temperature.** Conversely, Fig. 9 (right) shows that diversity scales nearly linearly with temperature, highlighting PRISM's ability to generate rich alternative sequences when encouraged to explore. Notably, diversity gains are not confined to "noise": even at moderate temperatures (e.g., $T = 0.7$), diversity is doubled relative to $T = 0.1$ while recovery remains $> 0.50$.

**Takeaway.** These results demonstrate that PRISM supports a *controllable accuracy–diversity trade-off* without collapsing at either extreme. By adjusting a single temperature parameter, users can shift seamlessly between high-fidelity recovery (for benchmarking) and diverse sequence generation (for design). This flexibility is rarely observed in prior inverse folding systems, which often either maximize recovery at the cost of trivial diversity or sacrifice fidelity under high-temperature sampling. The ablation therefore underscores PRISM's strength as not only an accurate but also a *versatile* framework for conditional protein design.

# N  USAGE OF LARGE LANGUAGE MODELS (LLMS) IN PAPER WRITING

LLMs were **used to polish the writing**. It **was not used** for retrieval, discovery, or research ideation.

