# OpenReview forum: "PRISM: Enhancing PRotein Inverse Folding through Fine- Grained Retrieval on Structure-Sequence Multimodal Representations"
_ICLR.cc/2026/Conference — ICLR 2026 Poster_

### Official Review · Reviewer_CkVi · 2025-10-27

**Soundness:** 3
**Presentation:** 3
**Contribution:** 3
**Rating:** 6
**Confidence:** 4

**Summary:**

The paper addresses the protein inverse folding problem: designing a sequence for a target 3D structure. The authors identify that existing methods lack mechanisms to reuse conserved, fine-grained structure-sequence patterns. They propose PRISM, a retrieval-augmented generation (RAG) framework that retrieves fine-grained "potential motif" representations from a vector database. These retrieved embeddings are integrated with global backbone context using a hybrid self-cross attention (MHSCA) decoder. The method is formulated as a latent-variable probabilistic model. Experiments across five benchmarks show that PRISM achieves new state-of-the-art results in sequence recovery and perplexity, while also improving structural foldability metrics.

**Strengths:**

The paper demonstrates strong empirical results, consistently outperforming existing methods, including its own base model (AIDO.Protein-IF), across five challenging benchmarks (CATH-4.2, TS50, TS500, CAMEO 2022, PDB date split).

The application of a retrieval-augmented generation (RAG) framework to inverse folding at a fine-grained, residue-level ("potential motif") is a novel contribution. This provides an explicit mechanism to reuse conserved local patterns, which is a limitation in prior work

The paper includes an extensive set of ablations that validate the core design choices, including the impact of the number of retrieved entries, the contribution of the hybrid decoder, the effect of aggregation depth, and the saturation of the vector database. The analysis of the recovery-diversity trade-off is also a valuable addition.

**Weaknesses:**

Presentation needs improvement. Some tables and figures overlap with text body, e.g., tab 3 and fig 3. Sec. 4.2 is missing. ‘Fig.’ and ‘Figure’ are both used in text body.

Some baselines, like SPDesign and VFN-IFE, that report higher metrics (on CATH), are not compared.

The training objective (Eq. 20) optimizes the parameters of the aggregation module ($\theta_Z$) and a structure encoder ($\theta_B$). However, the joint encoder $\mathcal{G}$ (identified as AIDO.Protein-IF) that produces the query embeddings $\hat{\mathcal{E}}^q$ and populates the database appears to be frozen, as the TopK retrieval step is non-differentiable. This prevents end-to-end training and means the model cannot learn to improve its representations for better retrieval; it can only learn to use the results from a fixed, and potentially suboptimal, retrieval set.

The "PRISM (str. enc.)" ablation (Tables 1, 2, 3) is vaguely described. The paper states this variant replaces the joint encoder with a "purely structure-based encoder (ProteinMPNN-CMLM)" and retrieval operates "only over structural embeddings". This seems to contradict the framework's core "multimodal representation" $\mathcal{E} = \mathcal{G}(B,S)$ (Eq. 5) and the definition of the database $D$. It is unclear how the multimodal vector database is constructed or queried in this "structure-only" setting.

**Questions:**

Let’s denote the training set as A, the RAG database as B. The paper compares models trained on A and augmented with B, with models trained on A. However, since the information in B is also used, could you compare models trained on A and augmented with B, with models trained on A+B?

Could you explain the independences for Eq. 3?

Is the joint encoder $\mathcal{G}$ (AIDO.Protein-IF) kept frozen during training? If so, have the authors considered methods to make the retrieval step differentiable (e.g., via relaxation) to enable end-to-end fine-tuning of the query encoder?

At inference, an "initial sequence estimate $\tilde{S}^q$" is required to generate the query embedding $\hat{\mathcal{E}}^q$. How sensitive is the model's final performance to the quality of this initial sequence? For instance, what is the drop in AAR if $\tilde{S}^q$ is generated by a weaker baseline like ProteinMPNN instead of AIDO.Protein-IF?

---

> ### Author Response · Authors · 2025-11-21
> **Thanks and response to Reviewer CkVi (part-1)**
>
> > Improvement requirements on presentation and text fixation
>
> Thank you very much for detecting the inconsistency and issues in the draft write up. We have now addressed them in the revised version.
>
> > Missing baselines for comparison
>
> Thank you very much for pointing this out. We would like to highlight that PRISM is base-estimator agnostic (as discussed earlier in our response to your previous point and shown in the table there). This means PRISM can in principle be paired with any stronger base estimator, including VFN-IFE or SPDesign, potentially yielding further improvements.
>
> Nevertheless, to have a comparison with stronger baselines we generated results and presented them in Table R4T1.
> In Table R4T1 below, we report PRISM’s performance under two different configurations and compare it with VFN-IF, VFN-IFE, LM-Design, and PiFold on  CATH-4.2. We follow the evaluation protocol of VFN-IFE [1] and FrameDiff [2] to compute the designability metrics *scTM>0.5 (%)* and *scRMSD<2 (%)*, which quantify the proportion of generated sequences whose ESMFold [3] predicted structures pass these structure-similarity filters.
>
> ### R4T1
> | Model | PPL $\downarrow$ | AAR (%) $\uparrow$ | scTM>0.5 (%) $\uparrow$ | scRMSD<2 (%)  $\uparrow$  |
> |--|--|--|--|--|
> | PiFold | 4.55 | 51.66 | 90.98 | 60.35 |
> | LM-Design | 4.41 | 54.41| 89.42 | 58.41 |
> | VFN-IF | 4.17 | 54.74 | 92.37 | 62.89 |
> | VFN-IFE | 3.36 | 62.67 | 93.29 | 64.16 |
> | PRISM (frozen $\mathcal{G}$) | 2.71 | 60.43 | 94.82 | **69.73** |
> | PRISM (fine-tuned $\mathcal{G}$, iteratively refined) | **2.65** | **63.33** | **95.36** | 69.55 |
>
> Here, to have a fair comparison with VFN-IFE we allow the joint-embedding module ($\mathcal{G}$) to be trainable in one setting. Additionally, we leverage a simple test-time output refinement strategy, denoted here by “iteratively refined”. For this the output sequence is iteratively fed back into the PRISM framework to improve the generation.
>
> As shown in R4T1, PRISM consistently yields similar or better performance. Regarding SPDesign, we have attempted to get clarity on the dataset size mismatch to have a fair comparison, and are awaiting for it. In order to ensure proper evaluation, we checked for the official codebase for clarification but could not locate one, so we used the SPDesign server (http://zhanglab-bioinf.com/SPDesign/) to request new predictions. We have submitted multiple queries and are currently waiting for the server’s outputs. We will be happy to provide the results upon resolution of these issues of the original work of SPDesign.
>
> [1] Mao, Weian, et al. "De novo Protein Design Using Geometric Vector Field Networks." The Twelfth International Conference on Learning Representations, 2024.
>
> [2] Yim, Jason, et al. "SE (3) diffusion model with application to protein backbone generation." Proceedings of the 40th International Conference on Machine Learning. 2023.
>
> [3] Lin, Zeming, et al. “Evolutionary-scale prediction of atomic-level protein structure with a language model”. Science, 2023.
>
> [4] Watson, Joseph L., et al. “Broadly applicable and accurate protein design by integrating structure prediction networks and diffusion generative models”. bioRxiv, 2022.
>
> > Demonstration on end-to-end trainability performance with joint embedding
>
> Originally we leveraged a “plug-and-play” frozen joint encoder. It was inline with minimizing the base training process of PRISM to be of dense retrieval with a token-level prediction loss (CE loss). This was primarily designed to reduce the training cost with fewer parameters, via a deterministic approximation.
> However, once we *relax* the assumption on $\mathcal{R}$, the KL term for $\mathcal{R}$ (i.e., $KL(q(\mathcal{R} |\cdot) || p(\mathcal{R} |\cdot))$) becomes non-zero. We can thus potentially minimize this by training the joint encoder (as suggested by the reviewer).
>
> To validate this, we conducted additional experiments in which we allowed the joint encoder ($\mathcal{G}$) to be trainable with the KL divergence loss ($KL(q(\mathcal{R} |\cdot) || p(\mathcal{R} |\cdot))$). Table R2T1 shows the efficacy of $\mathcal{G}$ being trainable. Note, due to limited resource and time we refrained from having hyperparameter optimization for the additional trainability test. We leave further hyperparameter optimization for accuracy improvement as part of future scope.
>
> ### R4T2
>
> | Model | PPL $\downarrow$ | AAR (%) $\uparrow$|
> |--|--|--|
> | LM-Design | 4.41 | 54.41|
> | DPLM | - | 54.54 |
> ||||
> | PiFold | 4.55 | 51.66 |
> | PRISM$\_{pifold}$ (frozen $\mathcal{G}$) | 2.72 | 60.09 |
> | PRISM$\_{pifold}$ (finetuned $\mathcal{G}$)  | **2.63** | **62.01** |
> ||||
> | AIDO.Protein-IF | 2.94 | 58.60 |
> | PRISM$\_{aido}$ (frozen $\mathcal{G}$) | 2.71 | 60.43 |
> | PRISM$\_{aido}$ (finetuned $\mathcal{G}$)  | **2.59** | **62.07** |
> ||||

---

> ### Author Response · Authors · 2025-11-21
> **Thanks and response to Reviewer CkVi (part-2)**
>
> >  It is unclear how the multimodal vector database is constructed or queried in this "structure-only" setting.
>
> Apologies for the potential confusion. PRISM (str. enc.) is developed for the purpose of ablation. Here the whole process is similar to the full PRISM pipeline, with the only difference is the embedding function here is only a structure encoder $F\_{\theta\_B}$ (ProteinMPNN-CMLM in our experiment) , i.e., here $\mathcal{E}=F\_{\theta\_B}(B)$, that does not take *any* sequence information. With this ablation we demonstrated the usefulness of PRISM under unimodal retrieval.
>
> > On comparison of models trained on training-set A and augmented with RAG database B, with models trained on A+B?
>
> We apologize for any confusion caused by our notation of the training set (A) and the retrieval database (B). In our framework, B is not an additional dataset beyond A, rather it is a vector-indexed representation of the same CATH-4.2 training split used to train the model. We have clarified this in the revised manuscript.
>
> > On explanation of the independences for Eq. 3
>
> Here we provide the explanation:
> 1. $\mathcal{E}$ \_||\_ $D \mid B$, because the joint-embedding  $\mathcal{E}$ of a protein does not depend on the vector-database $D$. Note that $\mathcal{E}$ is independent of $D$ even unconditionally in the DAG in Fig. 1, as we consider the crude embedding as a sample from the marginal $P(\mathcal{E}|B)$.
> 2. $\mathcal{R}$ \_||\_ $B \mid \mathcal{E}$, because the retrieval is conditionally independent of the backbone given the embeddings. The only path from $B$ to $\mathcal{R}$ is $B \rightarrow \mathcal{E} \rightarrow \mathcal{R}$. Conditioning on the non-collider $\mathcal{E}$ blocks this path.
> 3. {${\mathbf{Z},\mathbf{S}}$} \_||\_ $D \mid R$, because $\mathbf{Z}$ and $\mathbf{S}$ are conditionally independent of the vector database given the retrieved entities $\mathcal{R}$. The only edge from $D$ is $D \rightarrow \mathcal{R}$.  $\mathbf{Z}$ has incoming edges from $\mathcal{R}, B$, and $\mathcal{E}$. And  $\mathbf{S}$ has incoming edges from $\mathcal{R}, B, \mathcal{E}$, and  $\mathbf{Z}$. So both the paths $D \rightarrow \mathcal{R} \rightarrow \mathbf{Z}$ and $D \rightarrow \mathcal{R} \rightarrow \mathbf{S}$ are blocked when we condition on $\mathcal{R}$. There are no alternative paths from $D$ to $\mathbf{Z}$ or $\mathbf{S}$.

---

> ### Author Response · Authors · 2025-11-21
> **Thanks and response to Reviewer CkVi (part-3)**
>
> > On sensitivity of model’s final performance to the quality of initial seq.
>
>
> To investigate this, we first compare PRISM across three different base estimators: ProteinMPNN-CMLM (weak), PiFold (weak), and AIDO.Protein-IF (strong). The results below show that PRISM consistently improves over each base model, and remarkably, PRISM$\_{pmpnn}$ and PRISM$\_{pifold}$ (both using *single-iteration* refinement) already achieve performance comparable to the stronger AIDO.Protein-IF baseline, which itself uses iterative refinement.
>
> ### R4T3
> | Model | CATH 4.2 test PPL $\downarrow$ | CATH 4.2 test AAR (%) $\uparrow$ | TS50 PPL $\downarrow$ | TS50 AAR (%) $\uparrow$ | TS500 PPL $\downarrow$ | TS500 AAR (%) $\uparrow$ |
> |--|--|--|--|--|--|--|
> | ProteinMPNN-CMLM | 5.03 | 48.62 | 3.46 | 53.68 | 3.35 | 56.45 |
> | PRISM$\_{pmpnn}$ | 2.82 | 58.02 | 2.52 | 65.03 | 2.35 | 69.0 |
> ||||||||
> |PiFold | 4.55 | 51.66 | 3.86 | 58.72 | 3.44 | 60.42 |
> | PRISM$\_{pifold}$ | 2.72 | 60.09 | 2.44 | 66.56 | 2.26 | 71.16 |
> ||||||||
> |AIDO.Protein-IF (iterative) | 2.94 | 58.60 | 2.68 | 66.19 | 2.42 | 69.66 |
> | PRISM$\_{aido}$ | 2.71 | 60.43 | 2.43 | 67.92 | 2.27 | 70.53 |
>
> These results show that **the performance gap between PRISM variants is small, even though the base estimators differ significantly in quality**.
>
>
> To further quantify the potential dependency and sensitivity to the *initial sequence*, we stratified proteins by the *quality of the initial estimate*, then binned proteins by their PiFold AAR and measured AAR of generated sequences by PRISM. Table R1T4 shows consistent improvement with PRISM across different bin of initial sequences.
> ### R4T4
> | PiFold AAR Bin (%)   |   Number of proteins |   PiFold Mean AAR (%) |   PRISM$\_{pifold}$ Mean AAR (%) |   Δ (%)|
> |:----|-----|-----|----|----|
> | 0–20                 |           12 |                 15.89 |                         18.48 |                 2.59 |
> | 20–30                |           74 |                 25.97 |                         29.52 |                 3.55 |
> | 30–40                |          131 |                 35.51 |                         42.63 |                 7.12 |
> | 40–50                |          272 |                 45.95 |                         55.22 |                 9.27 |
> | 50–60                |          464 |                 54.88 |                         64.6  |                 9.72 |
> | 60–70                |          157 |                 62.79 |                         71.47 |                 8.68 |
> | 70–80                |           10 |                 73.57 |                         80.03 |                 6.46 |
>
> Here lower the bin value, lower the quality of the initial sequence.

---

> > ### Comment · Reviewer_CkVi · 2025-11-23
> >
> > Thank you for your response, my points are addressed. As the authors state that "PRISM is base-estimator agnostic", I suggest incorporating relevant protein sequence design approaches beyond inverse folding models into the related work section and discuss the potential of PRISM to combine with them in discussion/conclusion/appendix. Examples are:
> > SurfPro: Functional Protein Design Based on Continuous Surface
> > BC-Design: A Biochemistry-Aware Framework for Highly Accurate Inverse Protein Folding
> > SurfDesign: Effective Protein Design on Molecular Surfaces

---

> > > ### Author Response · Authors · 2025-11-24
> > > **Re: Thank you!**
> > >
> > > We thank the reviewer for their kind response and are glad to note that the rebuttal could address their concerns.
> > > We will incorporate the suggested "protein sequence design" approaches, to keep the related works broad, beyond inverse folding models. We also take note of the examples the reviewer suggested, will add them in the camera ready.

---

### Official Review · Reviewer_1DYG · 2025-10-30

**Soundness:** 3
**Presentation:** 3
**Contribution:** 3
**Rating:** 8
**Confidence:** 3

**Summary:**

The paper proposes a RAG system for the protein inverse folding task. It uses AIDO.Protein-IF as a joint protein sequence-structure encoder to generate residue-level embeddings for every protein in the knowledge base, which are then used to construct a RAG system. During inference, an initial guess of the protein sequence is first generated from a base estimator, then used to index and retrieve top-k candidates from the RAG. The paper claims SOTA performance of their approach on various benchmarks compared to existing baselines.

**Strengths:**

* The paper addresses the protein inverse folding problem, which is a core challenge in proteomics and biology.
* The residue-level RAG system allows flexible reuse of local structure and sequence patterns.
* The performance of the model is impressive on different benchmarks for its SOTA amino-acid recovery rate, perplexity and foldability.

**Weaknesses:**

* The model relies heavily on the AIDO.Protein-IF model in both RAG system construction and inference. The not-so-significant improvement in performance of the proposed method compared to AIDO.Protein-IF makes the entire work more like a second-stage refinement rather than a completely novel end-to-end framework.
* I suspect that the performance of the model might rely heavily on the quality of the initial guess from the base estimator AIDO.Protein-IF. This may be a particular problem when the retrieval system is at residue-level. The paper provides little further interpretation or ablation study on this matter.

**Questions:**

Please address my two major concerns in the “Weaknesses” section first. I will reassess after the rebuttal. Two other miscellaneous questions are as follows:
* The RAG system likely limits the generalization ability of the model on unseen protein backbones. Have you tested the performance of the model on novel or orphan proteins that are dissimilar to any known protein in the RAG?
* How much disk space do we need to store residue-level embeddings for every protein in the knowledge base? And what size of RAM did you use for experiments in section 4.6?

---

> ### Author Response · Authors · 2025-11-21
> **Thanks and response to Reviewer 1DYG (part-1)**
>
> > Heavy reliance on AIDO.Protein-IF model
>
> While AIDO.Protein-IF is indeed one option for both the base estimator and the joint embedding model, PRISM is not dependent on this choice, nor it is targeted to be a second-stage refinement of AIDO. Instead, PRISM is a general retrieval-augmented generation framework, highlighting the key insight that **inverse folding can benefit from explicit retrieval mechanisms that leverage the fine-grained diversity of known proteins**. Below, we provide empirical evidence demonstrating that PRISM’s design and performance are not tied to AIDO.Protein-IF.
>
> **1. PRISM is agnostic to the choice of base estimator.**
> Our proposed framework improves performance across weak, moderate, and strong base models, *showing that it is not tied to AIDO.Protein-IF or any specific initializer*. We conducted additional experiments to demonstrate this and summarize the new results in the Table R3T1.
>
> ### R3T1
> | Model | CATH 4.2 test PPL $\downarrow$ | CATH 4.2 test AAR (%) $\uparrow$ | TS50 PPL $\downarrow$ | TS50 AAR (%) $\uparrow$ | TS500 PPL $\downarrow$ | TS500 AAR (%) $\uparrow$ |
> |--|--|--|--|--|--|--|
> | ProteinMPNN-CMLM (base est.) | 5.03 | 48.62 | 3.46 | 53.68 | 3.35 | 56.45 |
> | PRISM$\_{pmpnn}$ | **2.82** | **58.02** | **2.52** | **65.03** | **2.35** | **69.0** |
> ||||||||
> | PiFol (base est.) | 4.55 | 51.66 | 3.86 | 58.72 | 3.44 | 60.42 |
> | PRISM$\_{pifold}$ | **2.72** | **60.09** | **2.44** | **66.56** | **2.26** | **71.16** |
> ||||||||
> |AIDO.Protein-IF (base est.)| 2.94 | 58.60 | 2.68 | 66.19 | 2.42 | 69.66 |
> | PRISM$\_{aido}$ | **2.71** | **60.43** | **2.43** | **67.92** | **2.27** | **70.53** |
>
> Here the subscripts $mpnn$, $pifold$ and $aido$ respectively denote whether ProteinMPNN-CMLM, PiFold, or AIDO.Protein-IF were used as the base estimator for PRISM.
>
> This clearly indicates that PRISM effectiveness is not tied to AIDO.Protein-IF or any specific initializer.
>
> **2. PRISM’s retrieval mechanism is agnostic to the choice of embedding model (either unimodal or multimodal).**
> To isolate the effect of retrieval itself, we constructed a *unimodal* PRISM variant using ProteinMPNN-CMLM as the embedding model, namely PRISM (str. enc.). We summarize the empirical results in the following table:
>
> ### R3T2
> | Model | CATH 4.2 test PPL $\downarrow$ | CATH 4.2 test AAR (%) $\uparrow$ | TS50 PPL $\downarrow$ | TS50 AAR (%) $\uparrow$ | TS500 PPL $\downarrow$ | TS500 AAR (%) $\uparrow$ | CAMEO2022 PPL $\downarrow$ | CAMEO2022 AAR (%) $\uparrow$ | PDB date PPL $\downarrow$ | PDB date AAR (%) $\uparrow$ |
> |--|--|--|--|--|--|--|--|--|--|--|
> | ProteinMPNN-CMLM | 5.03 | 48.62 | 3.46 | 53.68 | 3.35 | 56.45 | 3.62 | 50.14 | 3.42 | 52.98 |
> | PRISM (str. enc.) | 3.39 | 49.17 | 3.10 | 54.41 | 2.88 | 57.66 | 3.20 | 51.20 | 3.04 | 53.85 |
>
> Even *unimodal* retrieval offers measurable improvements over the base estimator.

---

> ### Author Response · Authors · 2025-11-21
> **Thanks and response to Reviewer 1DYG (part-2)**
>
> > On the doubt of performance of the model being heavily dependent on the base estimator
>
> To understand the impact of the “base estimator” quality in generating the final sequence we have conducted additional experiments across multiple base estimators and protein groups with widely varying initial sequence accuracies. Specifically, we evaluated PRISM when initialized by three base estimators of varying strengths: ProteinMPNN-CMLM (weak), PiFold (weak), and AIDO.Protein-IF (strong). In specific, as shown in Table R3T1, PRISM improves each of these estimators by a large margin. Notably, PRISM initialized from weaker estimators often achieves performance comparable to that with the stronger ones.
>
> To further quantify the potential dependency and sensitivity to the *initial sequence*, we stratified proteins by the *quality of the initial estimate*, then binned proteins by their PiFold AAR and measured AAR of generated sequences by PRISM. Table R3T3 shows consistent improvement with PRISM across different bin of initial sequences.
>
> ### R3T3
> | PiFold AAR Bin (%)   |   Number of proteins |   PiFold Mean AAR (%) |   PRISM$\_{pifold}$ (original $\mathcal{G}$, single-iter) Mean AAR (%) |   Δ (original $\mathcal{G}$, single-iter) |
> |:----|-----:|-----:|----:|----:|
> | 0–20                 |           12 |                 15.89 |                         18.48 |                 2.59 |
> | 20–30                |           74 |                 25.97 |                         29.52 |                 3.55 |
> | 30–40                |          131 |                 35.51 |                         42.63 |                 7.12 |
> | 40–50                |          272 |                 45.95 |                         55.22 |                 9.27 |
> | 50–60                |          464 |                 54.88 |                         64.6  |                 9.72 |
> | 60–70                |          157 |                 62.79 |                         71.47 |                 8.68 |
> | 70–80                |           10 |                 73.57 |                         80.03 |                 6.46 |
>
> Here lower the bin value, lower the quality of the initial sequence.
>
>
> > testing performance on novel or orphan proteins that are dissimilar to any known protein in the RAG
>
> We agree that evaluating retrieval-based methods solely on recovery of natural sequences does not directly assess their ability to generalize to novel or orphan backbones. To address this, we have conducted additional experiments on the **Orphan Proteins Dataset** (PNAS 2024) [1].
>
> Because our retrieval database (developed with CATH-4.2 train split) is derived from PDB, and all PDB sequences appear in at least one of the above databases, the Orphan Protein Dataset is **guaranteed to be non-overlapping** with both our training set and our retrieval index. We summarize the results in R3T4.
>
> ### R3T4
> | Model | AAR (%) $\uparrow$| sc-TM $\uparrow$ | RMSD $\downarrow$| pLDDT $\uparrow$|
> |--|--|--|--|--|
> | Native Sequence | 1 | 0.360 (0.449) | 3.61 (3.654) | 0.457 (0.535) |
> ||||||
> | AIDO.Protein-IF | 28.05 (30.22) | 0.369 (0.420) | 3.78 (3.615) | 0.538 (0.535) |
> | PRISM$\_{aido}$ | 28.33 (30.85) | 0.391 (0.437) | 3.92 (3.846) | 0.556 (0.531) |
> ||||||
> | ProteinMPNN-CMLM | 29.27 (31.24) | 0.359 (0.428) | 3.6 (3.712) | 0.569 (0.525) |
> | PRISM$\_{pmpnn}$ | 27.12 (30.76) | 0.404 (0.437) | 3.6 (3.587) | 0.533 (0.538) |
> ||||||
>
> **PRISM consistently improves over its respective base estimators** and achieves foldability scores close to the performance achievable by the native sequence, despite having no overlapping homologs in its retrieval database.
>
> Additionally, Table R3T4 shows that **PRISM may not simply memorize or copy natural proteins from the retrieval corpus**. Instead, it can meaningfully leverage retrieved structural context to enhance design **even when the target backbone is far outside the distribution of natural proteins**.
>
> [1] Single-sequence protein structure prediction by integrating protein language models. Proceedings of the National Academy of Sciences, 2024.
>
> > On disk space and RAM details for the embeddings and experiments of Sec 4.6, respectively
>
> Below we provide the exact resource requirements used in our experiments.
>
> - Disk storage for residue-level embeddings: 15.1 GB.
> - System RAM used: 256 GB (Most inference-time memory is handled on GPU, as the vector database is kept in GPU memory.)
> - GPU resources: Experiments in Sec. 4.6 were performed on 4 $\times$ NVIDIA A100 (40 GB each), for a total of 160 GB VRAM.
>
> We have included these details in Appendix B.1.

---

> > ### Comment · Reviewer_1DYG · 2025-11-26
> >
> > Thank you for the detailed response and the additional experimental results. All my concerns have been fully addressed, and I will stand by my strong recommendation to accept this paper. I suggest that the authors include these new results, especially the table concerning different PiFold AAR bins, in the main text of the paper.

---

### Official Review · Reviewer_3CLu · 2025-11-02

**Soundness:** 2
**Presentation:** 2
**Contribution:** 2
**Rating:** 2
**Confidence:** 3

**Summary:**

This paper proposes PRISM, a multimodal retrieval-augmented generation (RAG) framework for protein inverse folding. The key idea is to augment a generative inverse-folding model with fine-grained structure–sequence retrieval from a large protein database. Instead of relying solely on end-to-end transformer inference, PRISM retrieves localized structural motifs or fragments and conditions decoding through a hybrid self-cross-attention decoder that integrates both query (target structure) and retrieved (reference) contexts.

**Strengths:**

- The problem setup and latent-variable formulation provide theoretical grounding for the proposed method.
- The ablation studies and runtime analysis provide clear evidence of the effectiveness of the proposed method.

**Weaknesses:**

- The author did not provide the database construction process. What structures have been used in the retrieval? Even without exact overlap, similar folds from the same CATH family may leak local sequence priors, effectively making the task easier.
- Equation (13): The $p(Z \mid B, \mathcal{R} )$ should be $p(Z \mid B, \mathcal{R} , \mathcal{E})$, where $\mathcal{E}$ is the encoding of the retrieved fragments.
- The PGM derivation seems to prove something trivial. The actual training process is just a dense retrieval with some next-token prediction loss.

**Questions:**

- Some of the figures have overlaps with the text, and some tables are too small to read.
- Section 4.2 is empty.
- The figures look too busy to read. You can either extend it to full size or remove some details.

---

> ### Author Response · Authors · 2025-11-21
> **Thanks and response to Reviewer 3CLu**
>
> > On the clarification of database construction process: even similar folds may potentially leak local sequence priors
>
> We apologize for not properly describing the database construction process. Our retrieval database is constructed **from the training split of CATH-4.2**, which is widely used to train recent inverse folding models [4-6] (Sec. 3.4). It is curated to ensure that the train/validation/test partitions are disjoint at both the topology and homology levels [1]. Thus, the CATH-4.2 test split has no fold-level or family-level overlap with this training split.
>
> The other four evaluation sets (TS50, TS500, CAMEO2022, and the PDB date split) were independently curated to exclude overlap with CATH-4.2-train or with contemporary PDB entries at the time of release [2,3]. Hence, none of these benchmarks share domain-level homologs or fold-level relatives with the proteins used in our retrieval database.
>
> Taken together, these properties ensure that **no structure or sequence from any evaluation set appears in the retrieval index**, thereby preventing any form of data leakage *by construction*.
>
> For completeness, our retrieval database consists of all sequences in the CATH-4.2 training split (a total of 3,941,775 residues), where each residue corresponds to a database entry as defined in Sec. 3.4. Since we leverage this dataset to train our model as well, we additionally mask the query sequence itself to prevent trivial self-retrieval during training. We have added these details in Sec. 3.4 and Appendix B.2.
>
> [1] Cath–a hierarchic classification of protein domain structures. Structure, 1997.
>
> [2] Direct prediction of profiles of sequences compatible with a protein structure by neural networks with fragment-based local and energy-based nonlocal profiles. Proteins: Structure, Function, and Bioinformatics, 2014.
>
> [3] Generative flows on discrete state-spaces: Enabling multimodal flows with applications to protein co-design. International conference on machine learning, 2024.
>
> [4] Mixture of experts enable efficient and effective protein understanding and design. bioRxiv, 2024.
>
> [5] Robust deep learning-based protein sequence design using proteinmpnn. Science, 2022.
>
> [6] Diffusion language models are versatile protein learners. International conference on machine learning, 2025.
>
> > Correction in Eq. 13
>
> Thank you for pointing this out. We have corrected the manuscript by changing $p(\mathbf{Z} \mid B, \mathcal{R})$ to $p(\mathbf{Z} \mid B, \mathcal{R}, \mathcal{E})$; now it is consistent with Eq. 3.
>
> > The actual training process is just a dense retrieval with some next-token prediction loss.
>
> Indeed the original training process of PRISM is a form of dense retrieval with a token-level prediction loss (CE loss). This was primarily designed to reduce the training cost with fewer parameters, via a deterministic approximation.
>
> However, the PGM derivation remains useful and can potentially yield non-trivial minimization objectives. For example, in deterministic assumptions on $\mathcal{E}$, $\mathcal{R}$, and $\mathbf{Z}$, the KL in Eq. (31) is zero yielding only the CE loss. However, once we *relax* the assumption on $\mathcal{R}$, the KL term for $\mathcal{R}$ (i.e., $KL(q(\mathcal{R} |\cdot) || p(\mathcal{R} |\cdot))$) becomes non-zero. We can thus potentially minimize this by training the joint encoder.
> To validate this, we conducted additional experiments in which we allowed the joint encoder ($\mathcal{G}$) to be trainable with this KL divergence loss. Table R2T1 shows the efficacy of $\mathcal{G}$ being trainable.
>
> ### R2T1
>
> | Model | PPL $\downarrow$ | AAR (%) $\uparrow$|
> |--|--|--|
> | LM-Design | 4.41 | 54.41|
> | DPLM | - | 54.54 |
> ||||
> | PiFold | 4.55 | 51.66 |
> | PRISM$\_{pifold}$ (frozen $\mathcal{G}$) | 2.72 | 60.09 |
> | PRISM$\_{pifold}$ (finetuned $\mathcal{G}$)  | **2.63** | **62.01** |
> ||||
> | AIDO.Protein-IF | Yes | 2.94 | 58.60 |
> | PRISM$\_{aido}$ (frozen $\mathcal{G}$) | 2.71 | 60.43 |
> | PRISM$\_{aido}$ (finetuned $\mathcal{G}$)  | **2.59** | **62.07** |
> ||||
>
> > On formatting issues: overlapped fig, empty section, too busy fig
>
> We thank the reviewer for pointing out these presentation issues. We have carefully revised the manuscript to address all three concerns.

---

> > ### Comment · Reviewer_3CLu · 2025-11-26
> >
> > Thanks for the clarifications on the data setup. Those points are clear to me now.  My only remaining concern is about the role of the PGM.
> >
> > In the deterministic setting used in the main paper (i.e., $E=f(B)$, $R=\mathrm{TopK}(E;D)$, $Z=A(E,B,R)$), the latent model $p(S,E,R,Z \mid B,D) = p(E\mid B)\cdotp(R\mid E,D)\cdotp(Z\mid R,E,B)\cdotp(S\mid Z,R,E,B)$ effectively collapses to maximizing $\log p_\theta(S\mid B,D)$ via standard cross-entropy.
> >
> > The KL term you introduce in the rebuttal seems to come from a generic variational retrieval $q(R\mid E)$ vs. prior $p(R\mid E)$, which does not really depend on this specific factorization. So I’m still not fully convinced what additional, functional constraint the proposed PGM is imposing on the actual trained model.

---

> ### Author Response · Authors · 2025-11-27
> **Re: Response to reviewer 3CLu**
>
> We thank reviewer 3CLu for the follow-up and for confirming that the data-splitting and leakage issues are now clear. We appreciate your continued engagement with the latent-variable formulation. We now intend to address your remaining concern and hope to receive your support in the final score.
>
> > On the deterministic setting.
>
> We agree with your observation that, under the deterministic instantiation used in the main paper, the training objective of the latent model *indeed* reduces to maximizing the standard $\log p_\theta(\mathbf{S}\mid B,D)$ (Eq. 20, App. F, Prop. 2). In other words, in this deterministic limit the PGM does not impose any *additional* optimization constraint beyond cross-entropy loss. We will make this more explicit in the revised draft.
>
> > What the PGM is intended to contribute.
>
> Our intention with the PGM is *not to claim* that the deterministic training objective is fundamentally different from cross-entropy, but rather to: (i) give a principled generative interpretation of the representation-retrieval-attribution-emission pipeline (Fig. 1-2), and (ii) show that the *same* cross-entropy loss can potentially be viewed as a tight prior-Jensen lower bound on a more general stochastic latent-variable model (Eq. 27-36). Concretely, App. F shows that when $p(\mathcal{E},\mathcal{R},\mathbf{Z}\mid B,D)$ degenerates to Dirac masses, the negative log-likelihood of the latent model equals the usual per-residue cross-entropy, and also upper-bounds the NLL of the corresponding stochastic model (Eq. 34); this is a functional connection we intended to establish.
>
> > Relation of the KL term to the factorization.
>
> We agree that the KL divergence term we discussed in the rebuttal takes the form of a "generic" variational retrieval regularizer between an amortized posterior $q(\mathcal{R}\mid\mathbf{S},\mathcal{E},D)$ and a prior $p(\mathcal{R}\mid \mathcal{E},D)$. What the *factorization* buys us here is that the ***prior* is exactly the retrieval kernel $p(\mathcal{R}\mid \mathcal{E},D)$** from the PGM (Sec. 3.5, App. E-F), and the KL term arises by applying the *standard* ELBO decomposition on Eq. 31 while keeping the other latents $\mathcal{E}$ and $\mathbf{Z}$ deterministic. Specifically, under factorization  of $q(\mathcal{E},\mathcal{R},\mathbf{Z}\mid \mathbf{S},B,D)$ and $p(\mathcal{E},\mathcal{R},\mathbf{Z}\mid B,D)$, the ELBO from Eq. 31 can be represented as follows,
>
> $\mathcal{L}\_{{ELBO}}(q) =\mathbb{E}\_{q}\big[\log p(\mathbf{S}\mid \cdot)\big]
> -\mathrm{KL}\big(q(\mathcal{E}\mid \mathbf{S},B) \mid\mid p(\mathcal{E}\mid B)\big)
> -\mathbb{E}\_{q}\big[\mathrm{KL}\big(q(\mathcal{R}\mid \mathbf{S},\mathcal{E},D) \mid\mid p(\mathcal{R}\mid \mathcal{E},D)\big)\big]
> -\mathbb{E}\_{q}\big[\mathrm{KL}\big(q(\mathbf{Z}\mid \mathbf{S},\mathcal{R},\mathcal{E},B) \mid\mid p(\mathbf{Z}\mid \mathcal{R},\mathcal{E},B)\big)\big]$
> $~~~~~~~~~~~~~~~~=\mathbb{E}\_{q}\big[\log p(\mathbf{S}\mid \cdot)\big]
> -0
> -\mathbb{E}\_{q}\big[\mathrm{KL}\big(q(\mathcal{R}\mid \mathbf{S},\mathcal{E},D) \mid\mid p(\mathcal{R}\mid \mathcal{E},D)\big)\big]
> -0$
> $~~~~~~~~~~~~~~~~=\mathbb{E}\_{q}\big[\log p(\mathbf{S}\mid \cdot)\big]-\mathbb{E}\_{q}\big[\mathrm{KL}\big(q(\mathcal{R}\mid \mathbf{S},\mathcal{E},D) \mid\mid p(\mathcal{R}\mid \mathcal{E},D)\big)\big].$
>
> In this spirit, the experimental evaluations with the additional objective of minimizing the KL divergence loss can be viewed as a variant of PRISM with relaxed assumption - it shows that if we relax the deterministic assumption on the retrieval prior, we can optimize a well-defined variational bound.

---

> > ### Comment · Reviewer_3CLu · 2025-11-27
> >
> > Thank you for your prompt clarification.
> >
> > Since the authors argue that the PGM mainly functions as a 'principled interpretation' and a 'functional connection' rather than an active optimization mechanism in the main results, the manuscript should be revised to reflect this fact. Additionally, the heavy mathematical derivation of the PGM currently dominates the Method section, potentially misleading readers into thinking it is the active training objective.
> >
> > I strongly suggest moving the detailed PGM derivation to the Appendix and simplifying Section 3 to focus on the actual deterministic architecture that yields the reported performance.

---

### Official Review · Reviewer_5aXk · 2025-11-04

**Soundness:** 3
**Presentation:** 3
**Contribution:** 3
**Rating:** 6
**Confidence:** 4

**Summary:**

The paper introduces PRISM, a novel framework for protein inverse folding.
The authors identify a key limitation in existing deep learning models: they lack an explicit mechanism to reuse conserved, fine-grained structure-sequence patterns (e.g., motifs) found in known proteins. To address this, PRISM is proposed as a multimodal retrieval-augmented generation (RAG) framework.
The core idea is to supplement end-to-end generation with a memory-based approach. The method works in several steps:
(1) construct a vector database by encoding known protein structures and sequences into contextualized representations.
(2) given a target 3D structure, leverage a base model to generate an initial sequence estimate, which is used to retrieve the vector database to obtain the most similar potential motifs.
(3) a novel hybrid self-cross attention (MHSCA) decoder then integrates the backbone information with the retrieved motifs to generate the sequence.
The method is theoretically grounded as a latent-variable probabilistic model. Empirically, PRISM improves upon all baselines in both sequence accuracy (AAR) and structural foldability (RMSD and TM-score) across five major benchmarks (CATH-4.2, TS50, TS500, CAMEO 2022, and PDB date split). Extensive and thorough ablation and analysis demonstrate the effectiveness of the proposed modules.

**Strengths:**

1. The core idea of explicitly retrieving fine-grained, residue-level motifs is an intuitive and novel solution to reuse the conserved local patterns which are central to protein function.
2. The method demonstrates clear and consistent improvement over existing models across all five evaluation benchmarks. This improvement is shown not only in sequence recovery metrics but also in more practical foldability scores.
3. The authors show that the significant improvement brought by retrieval-based method are achieved with a negligible runtime overhead (only a 14.3% compared to the base estimator), making PRISM a practical and scalable solution.
4. The authors provide extensive ablation studies to justify their design choices. They successfully prove the contribution of the retrieval mechanism and improvement over varying sequence lengths, optimize the number of retrieved entries, validate the hybrid MHSCA decoder design, and analyze the effect of decoder depth.

**Weaknesses:**

1. The main results in Table 1 should include structure-related metrics (e.g., scTM and RMSD) instead of relying solely on AAR and PPL for evaluation. Although scTM and RMSD are assessed in Table 4, this comparison is limited to DPLM2-3B and AIDO and lacks a broader set of baselines.
2. Several critical details are missing from the main body text:
- What protein library was the vector database built upon?
- In Figure 2, do the structure embedding and joint embedding originate from the same model? Are they kept fixed during training?
- A new term, "base estimator," appears in line 315. I speculate it is used to estimate the initial sequence during inference. If so, the authors should explicitly clarify it.
3. The sampling process relies on the "base estimator" to generate an initial sequence, which is then used for subsequent retrieval. This raises a key question: does the quality of this initial sequence significantly impact the final generated sequence? The authors have not provided an analysis of this potential dependency.
4. The retrieval-based generation method may cause the generated protein sequences to be overly similar to natural proteins (as suggested by the high AAR of ~70% in Table 2), which will limit the novel protein design. Furthermore, it's uncertain if the method's performance will drop significantly when used on a novel backbone that not similar to any entries in the database.
5. Minor issue: there is a formatting error where Table 3 overlaps with the main text.

**Questions:**

See above weaknesses.

---

> ### Author Response · Authors · 2025-11-21
> **Thanks and response to Reviewer 5aXk (part-1)**
>
> > Provide additional structure related metrics in Table 1 over more baselines
>
> Following the reviewer’s suggestion, we have now computed **sc-TM**, **RMSD**, and **pLDDT** across a broader set of baselines, using the widely adopted ESMFold structure predictor [1]. As shown in R1T1, PRISM consistently improves or matches foldability metrics across all base estimators, further supporting its effectiveness.
> ### R1T1
> | Model | RMSD  $\downarrow$ | sc-TM $\uparrow$ | pLDDT $\uparrow$|
> |--|--|--|--|
> | Native Sequence | 1.42 | 0.931 | 0.842 |
> |||||
> | ProteinMPNN-CMLM | 1.64 | 0.910 | 0.812 |
> | PiFold | 1.64 | 0.911 | 0.816 |
> | AIDO.Protein-IF | 1.58 | 0.912 | 0.831 |
> | PRISM  | **1.56** | **0.913** | **0.835** |
>
> Here we also provide foldability scores with native sequence for reference. We have integrated these results into Tab.1 in the main text.
>
> [1] Evolutionary-scale prediction of atomic-level protein structure with a language model, Science 2023.
>
> > Missing critical details
>
> We apologize for not clearly highlighting some of the critical details. Our retrieval database is constructed **from the training split of CATH-4.2**, which is widely used to train recent inverse folding models [4-6] (Sec. 3.4). It is curated to ensure that the train/validation/test partitions are disjoint at both the topology and homology levels [1]. Thus, the CATH-4.2 test split has no fold-level or family-level overlap with this training split.
>
> The other four evaluation sets (TS50, TS500, CAMEO2022, and the PDB date split) were independently curated to exclude overlap with CATH-4.2-train or with contemporary PDB entries at the time of release [2,3]. Hence, none of these benchmarks share domain-level homologs or fold-level relatives with the proteins used in our retrieval database.
>
> Taken together, these properties ensure that **no structure or sequence from any evaluation set appears in the retrieval index**, thereby preventing any form of data leakage *by construction*.
>
> For completeness, our retrieval database consists of all sequences in the CATH-4.2 training split (a total of 3,941,775 residues), where each residue corresponds to a database entry as defined in Sec. 3.4. Since we leverage this dataset to train our model as well, we additionally mask the query sequence itself to prevent trivial self-retrieval during training. We have added these details in Sec. 3.4 and Appendix B.2.
>
> [1] Cath–a hierarchic classification of protein domain structures. Structure, 1997.
>
> [2] Direct prediction of profiles of sequences compatible with a protein structure by neural networks with fragment-based local and energy-based nonlocal profiles. Proteins: Structure, Function, and Bioinformatics, 2014.
>
> [3] Generative flows on discrete state-spaces: Enabling multimodal flows with applications to protein co-design. International conference on machine learning, 2024.
>
> [4] Mixture of experts enable efficient and effective protein understanding and design. bioRxiv, 2024.
>
> [5] Robust deep learning-based protein sequence design using proteinmpnn. Science, 2022.
>
> [6] Diffusion language models are versatile protein learners. International conference on machine learning, 2025.
>
> >  Clarification on structured and joint embedding origination
>
> The *structure embedding* originates from ProteinMPNN-CMLM, a non-autoregressive reimplementation of the ProteinMPNN, whereas the *joint embedding* originates from AIDO.Protein-IF (Figure 2 in the main text, and Appendix G.1).
>
> >  Clarification on structured and joint embedding trainability
>
> The *joint encoder* $\mathcal{G}$ *is kept frozen* (16.2B parameters) during our main experiments, primarily because fine-tuning may not often be supported at limited resource hardware. In contrast, the *structure encoder is fine-tuned*  (1.7M parameters) jointly with the decoder, demonstrating strong performance gains at limited fine-tuning parameter cost. Note that AIDO.Protein-IF also uses ProteinMPNN-CMLM as its structure encoder, but it is not parameter-shared with PRISM's in the main experiments, ensuring $\mathcal{G}$ remains truly frozen. In Figure 2, `Frozen` and `Trainable` markers identify this trainability.
>
> Additionally, we conducted ablation on finetuning $\mathcal{G}$ with low-rank adaptation, where we do allow parameter-sharing of the structure encoders for efficiency. The results are summarized below:
>
> ### R1T2
> | Model | PPL $\downarrow$ | AAR (%) $\uparrow$|
> |--|--|--|
> | LM-Design | 4.41 | 54.41|
> | DPLM |-| 54.54 |
> | PiFold | 4.55 | 51.66 |
> | PRISM$\_{pifold}$ (frozen $\mathcal{G}$) | 2.72 | 60.09 |
> | PRISM$\_{pifold}$ (finetuned $\mathcal{G}$)  | **2.63** | **62.01** |
>
> Here, the subscript ${pifold}$ indicates PiFold as the *base estimator used for PRISM*. Notably, due to limited rebuttal time, we did not perform any hyperparameter tuning to allow the $\mathcal{G}$ to be trainable. Thus, the performance with $\mathcal{G}$ trainable can potentially improve with proper hyperparameter tuning.

---

> ### Author Response · Authors · 2025-11-21
> **Thanks and response to Reviewer 5aXk (part-2)**
>
> > Clarification on the “base estimator” in L315
>
> Indeed, the term “base estimator’’ refers to the off-the-shelf inverse-folding model used to generate the initial sequence $\tilde{S}^q$ during inference. We have updated the manuscript to explicitly introduce and define this term at its first appearance, ensuring that its role is immediately clear.
>
> > On the quality of “base estimator" driven  initial sequence impacting the final generated sequence
>
> To understand the impact of the “base estimator” quality in generating the final sequence we have now conducted additional experiments across multiple base estimators and protein groups with widely varying initial sequence accuracies. Specifically, we evaluated PRISM when initialized by three base estimators of varying strengths: ProteinMPNN-CMLM (weak), PiFold (weak), and AIDO.Protein-IF (strong). As shown in Table R1T3, PRISM improves each of these estimators by a large margin, and notably, PRISM initialized from weaker estimators often achieves performance comparable to that with the stronger ones.
>
> ### R1T3
> | Model | CATH 4.2 test PPL $\downarrow$ | CATH 4.2 test AAR (%) $\uparrow$ | TS50 PPL $\downarrow$ | TS50 AAR (%) $\uparrow$ | TS500 PPL $\downarrow$ | TS500 AAR (%) $\uparrow$ |
> |--|--|--|--|--|--|--|
> | ProteinMPNN-CMLM | 5.03 | 48.62 | 3.46 | 53.68 | 3.35 | 56.45 |
> | PRISM$\_{pmpnn}$ | **2.82** | **58.02** | **2.52** | **65.03** | **2.35** | **69.0** |
> ||||||||
> |PiFold | 4.55 | 51.66 | 3.86 | 58.72 | 3.44 | 60.42 |
> | PRISM$\_{pifold}$ | **2.72** | **60.09** | **2.44** | **66.56** | **2.26** | **71.16** |
> ||||||||
> |AIDO.Protein-IF | 2.94 | 58.60 | 2.68 | 66.19 | 2.42 | 69.66 |
> | PRISM$\_{aido}$ | **2.71** | **60.43** | **2.43** | **67.92** | **2.27** | **70.53** |
>
> > On the limit of novel protein design: overly similar protein sequence generation due to retrieval-based approach
>
> We agree that evaluating retrieval-based methods solely on recovery of natural sequences does not directly assess their ability to generalize to novel or orphan backbones. To address this, we have conducted additional experiments on the **Orphan Proteins Dataset** (PNAS 2024) [1].
>
> Because our retrieval database (developed with CATH-4.2 train split) is derived from PDB, and all PDB sequences appear in at least one of the above databases, the Orphan Protein Dataset is **guaranteed to be non-overlapping** with both our training set and our retrieval index. We summarize the results in R1T4.
>
> ### R1T4
> | Model | AAR (%) $\uparrow$| sc-TM $\uparrow$ | RMSD $\downarrow$| pLDDT $\uparrow$|
> |--|--|--|--|--|
> | Native Sequence | 1 | 0.360 (0.449) | 3.61 (3.654) | 0.457 (0.535) |
> ||||||
> | AIDO.Protein-IF | 28.05 (30.22) | 0.369 (0.420) | 3.78 (3.615) | 0.538 (0.535) |
> | PRISM$\_{aido}$ | 28.33 (30.85) | 0.391 (0.437) | 3.92 (3.846) | 0.556 (0.531) |
> ||||||
> | ProteinMPNN-CMLM | 29.27 (31.24) | 0.359 (0.428) | 3.6 (3.712) | 0.569 (0.525) |
> | PRISM$\_{pmpnn}$ | 27.12 (30.76) | 0.404 (0.437) | 3.6 (3.587) | 0.533 (0.538) |
> ||||||
>
> **PRISM consistently improves over its respective base estimators** and achieves foldability scores close to the performance achievable by the native sequence, despite having no overlapping homologs in its retrieval database.
>
> Additionally, Table R1T4 shows that **PRISM may not simply memorize or copy natural proteins from the retrieval corpus**. Instead, it can meaningfully leverage retrieved structural context to enhance design **even when the target backbone is far outside the distribution of natural proteins**.
>
> [1] Single-sequence protein structure prediction by integrating protein language models. Proceedings of the National Academy of Sciences, 2024
>
> > Minor: formatting error
>
> We have corrected the formatting errors in the updated main draft.

---

> ### Author Response · Authors · 2025-11-27
> **Request from the authors**
>
> Dear reviewer 5aXk,
>
> Thank you for your detailed and insightful review of our draft. Taking this into account we have updated the manuscript while we tried to address majority of your concerns through empirical evidence. We would hope to receive your feedback and remark on the rebuttal response we prepared.
>
> Authors

---

### Meta-Review · Area_Chair_B2Bi · 2026-01-06

**Summary:**

Designing protein sequences that fold into a specified 3D structure is a fundamental task in computational biology. This study presents **PRISM**, a retrieval-augmented generation (RAG) framework applied at the fine-grained, residue-level motif scale. PRISM explicitly addresses a key limitation of existing inverse folding models—their lack of mechanisms to reuse conserved local structure-sequence patterns—and provides a flexible solution for motif reuse. Extensive experiments demonstrate that this method achieves consistent state-of-the-art (SOTA) performance across all baseline models. Also, the method is formulated as a latent-variable probabilistic model, providing theoretical grounding.

**Key concerns from the reviewers**

**1. Ambiguity of retrieval database and potential generalization limitations (raised by all the Reviewers)**

Critical details regarding the vector database construction process are not provided. Key unresolved questions include: whether the vector database contains similar folds from the same CATH family, which could lead to potential data leakage; and whether the retrieval-based method can be generalized to novel/orphan backbones with no homologous entries in the database. Additionally, more detailed information should be provided on the disk space required for storing residue-level embeddings and the RAM specifications used in the experiments.

**2. PRISM is highly relied on initial sequence or based model (raised by Reviewer 5aXk, 1DYG, and CkVi) **

The quality of the initial sequence generated by the "base estimator" (AIDO.Protein-IF) is critical for subsequent retrieval, yet no analysis of this dependency has been conducted or reported. Furthermore, PRISM relies heavily on AIDO.Protein-IF for both database construction and inference. The marginal performance improvement observed over AIDO.Protein-IF makes the work appear more like a second-stage refinement of the base model rather than a completely novel, end-to-end framework.


**3. Insufficient experiments (raised by Reviewer 5aXk, CkVi )**
The main results table relies solely on sequence-level metrics, with no inclusion of key structural foldability metrics. Additionally, the evaluation lacks comparisons against several important strong baselines that are relevant to the task.

**4. Theoretical & Training Limitations (by Reviewer 3CLu, CkVi)**

The joint encoder (AIDO.Protein-IF) used for database construction and query generation is frozen during training (due to non-differentiable TopK retrieval). This prevents end-to-end training and limits the model’s ability to improve retrieval representations. The probabilistic graphical model (PGM) derivation is considered trivial.

**Reviewer Concerns:**

I believe the authors have successfully addressed the major concerns by presenting an extensive set of new experimental results and analyses.

**Reviewer Scores:**

I suggest that Reviewers 5aXk, 1DYG, and CkVi would maintain their original scores, or even increase them, given the comprehensive revisions. However, I remain unclear as to why Reviewer 3CLu assigned a score of only 2.

---

### Decision · Program_Chairs · 2026-01-26

Accept (Poster)